# Exploring Jacobian Inexactness in Second-Order Methods for Variational Inequalities: Lower Bounds, Optimal Algorithms and Quasi-Newton Approximations

**Artem Agafonov[1, 2][*], Petr Ostroukhov[1, 2], Roman Mozhaev[2], Konstantin Yakovlev[2], Eduard Gorbunov[1], Martin Takáč[1], Alexander Gasnikov[3,2,4], Dmitry Kamzolov[1]**

[1] Mohamed bin Zayed University of Artificial Intelligence (MBZUAI), Abu Dhabi, UAE
[2] Moscow Institute of Physics and Technology (MIPT), Moscow, Russia
[3] Innopolis University, Innopolis, Russia
[4] Skoltech, Moscow, Russia

## Abstract

Variational inequalities represent a broad class of problems, including minimization and min-max problems, commonly found in machine learning. Existing second-order and high-order methods for variational inequalities require precise computation of derivatives, often resulting in prohibitively high iteration costs. In this work, we study the impact of Jacobian inaccuracy on second-order methods. For the smooth and monotone case, we establish a lower bound with explicit dependence on the level of Jacobian inaccuracy and propose an optimal algorithm for this key setting. When derivatives are exact, our method converges at the same rate as exact optimal second-order methods. To reduce the cost of solving the auxiliary problem, which arises in all high-order methods with global convergence, we introduce several Quasi-Newton approximations. Our method with Quasi-Newton updates achieves a global sublinear convergence rate. We extend our approach with a tensor generalization for inexact high-order derivatives and support the theory with experiments.

## 1 Introduction

In this paper, we primarily address the problem of solving the Minty Variational Inequality (MVI) [77, 17]. Given a continuous operator $F : \mathcal{X} \to \mathbb{R}^d$, where $\mathcal{X} \subseteq \mathbb{R}^d$ is a closed bounded convex subset with a diameter $D = \max_{x,y \in \mathcal{X}} \|x - y\|$, the objective is to find a point $x^* \in \mathcal{X}$ such that

$$\langle F(x), x - x^* \rangle \geq 0, \quad \text{for all } x \in \mathcal{X}. \tag{1}$$

The solution to (1) is referred to as a weak solution of the Variational Inequality (VI) [37]. In contrast, the Stampacchia variational inequality problem [48] consists in finding a point $x^* \in \mathcal{X}$ such that

$$\langle F(x^*), x - x^* \rangle \geq 0, \quad \text{for all } x \in \mathcal{X}. \tag{2}$$

This solution is often called a strong solution to the variational inequality. When the operator $F$ is both continuous and monotone, the weak and strong solutions are equivalent [37].

---

[*]Contact details: Artem Agafonov - `agafonov.ad@phystech.edu`; Petr Ostroukhov - `postroukhov12@gmail.com`; Roman Mozhaev - `mozhaev.rm@phystech.edu`; Konstantin Yakovlev - `iakovlev.kd@phystech.edu`; Eduard Gorbunov - `eduard.gorbunov@mbzuai.ac.ae`; Martin Takáč - `takac.MT@gmail.com`; Alexander Gasnikov - `gasnikov@yandex.ru`; Dmitry Kamzolov - `kamzolov.opt@gmail.com`.

38th Conference on Neural Information Processing Systems (NeurIPS 2024).

**Assumption 1.1** *The operator $F(x)$ is called monotone, if*
$$\langle F(x) - F(y), x - y \rangle \geq 0, \quad \text{for all } x, y \in \mathcal{X}. \tag{3}$$

Another useful assumption is $L_1$-smoothness.

**Assumption 1.2** *The operator $F(x)$ is $L_1$-smooth, if it has Lipschitz-continuous first-order derivative*
$$\|\nabla F(x) - \nabla F(y)\|_{\text{op}} \leq L_1 \|x - y\|, \quad \text{for all } x, y \in \mathcal{X}.$$

**First-order methods.** Variational inequalities encompass a wide range of problems, including minimization [19, 90, 93, 15], min-max problems, Nash equilibrium, differential equations, and others [37, 16]. The extensive research on VI methods dates back several decades, with a notable breakthrough in the 1970s—the development of the Extragradient method [62, 6]. Subsequently, it was demonstrated that this method achieves global convergence of $O\left(\varepsilon^{-1}\right)$ [80, 43], matching the convergence rates of other first-order[2] methods such as optimistic gradient [92, 78, 63, 44], forward-backward splitting [95], and dual extrapolation [81]. These first-order methods collectively exhibit optimal convergence [89].

**Second-order and high-order methods.** To achieve further notable acceleration of methods for VIs, one can leverage information about higher-order derivatives. For instance, simply incorporating first-order derivatives (Jacobian) can significantly enhance the convergence speed of the method. Following recent advancements in second-order and high-order methods with global rates for minimization [85, 82, 10, 79, 83, 46, 58, 14, 51, 64, 22], several high-order methods for VIs have been proposed [21, 68, 52, 88, 70, 84, 72, 5]. However, all these methods involve a line-search procedure, resulting in $\tilde{O}\left(\varepsilon^{-2/3}\right)$ convergence for the case of first-order information (Jacobians). Recent works [69, 2] propose methods with improved rates $O\left(\varepsilon^{-2/3}\right)$ and establish the lower bound $\Omega\left(\varepsilon^{-2/3}\right)$, rendering these algorithms optimal.

**Jacobian's approximation.** In the last decade, VIs found new applications in machine learning. There are many problems that could not be reduced to minimization, including reinforcement learning [87, 55], adversarial training [75], GANs [42, 31, 41, 76, 28, 67, 91], classical learning tasks in supervised learning [56, 9], unsupervised learning [98, 8], image denoising [35, 27], robust optimization [13]. Applying second-order methods, without even mentioning high-order ones, described in a previous paragraph to machine learning problems could be a challenging task. Although these methods may theoretically converge faster, computing exact Jacobians and the per-iteration costs can be expensive. Therefore, it seems natural to introduce inexact approximations of the first-order derivatives. In the context of minimization, several works with inexact Hessians were introduced for both convex [40, 3, 7, 4] and nonconvex [24, 25, 23, 61, 99, 94, 74, 11, 12, 34] problems. Regarding VIs, Quasi-Newton (QN) methods [73, 14, 57] can be highlighted, though they, unfortunately, achieve only local convergence in the strongly monotone case [18, 38]. These methods are relatively less advanced for VIs compared to their counterparts in the field of minimization, where they are considered classics in optimization due to their effectiveness and practicality [86]. Modern research on QN approximations for minimization includes methods that exhibit global convergence [57, 54, 53]. A recent work [71] introduces the Newton-MinMax method for convex-concave unconstrained min-max optimization problems, demonstrating an optimal rate under special assumptions on the accuracy of the Jacobian approximation. However, the field of VIs lacks globally convergent inexact second-order methods with an explicit dependence on the accuracy of the Jacobian. This raises several natural questions:

*What are the lower bounds for methods with inexact Jacobians?*
*Can we construct an optimal method with inexact first-order information?*
*What is the proper way to approximate the Jacobian to ensure global convergence and reduce the iteration complexity?*

In our work, we attempt to answer these questions in a systematic manner.

---

[2]For clarity, let us note that VI methods that use only the information about the operator itself are commonly referred to as first-order methods. Methods that also use information about the Jacobian of the operator (the first-order derivative) are known as second-order methods. This somewhat contradictory notation stems from applications to minimization problems, where the operator is the gradient (i.e., the first-order derivative), and the Jacobian is the Hessian (i.e., the second-order derivative).

**Optimality measure.** Most of our results are stated for the monotone setting (Assumption 1.1). In this context, the optimality of a point $\hat{x} \in \mathcal{X}$ is typically measured by a gap function $\text{GAP}(\cdot) : \mathcal{X} \to \mathbb{R}_+$ [95, 80, 81, 78, 69], defined by

$$\text{GAP}(\hat{x}) = \sup_{x \in \mathcal{X}} \langle F(x), \hat{x} - x \rangle \leq \varepsilon, \tag{4}$$

where $\varepsilon \geq 0$ is the accuracy of solution. The boundedness of $\mathcal{X}$ and the existence of a strong solution ensure that the gap function is well-defined. If $\varepsilon = 0$, we get by (1) that $\hat{x}$ is a weak solution of VI. We explore the performance of the proposed algorithm in scenarios involving nonmonotone operators $F$. In such cases, it is essential to assume that the operator satisfies the Minty condition to ensure that the problem is computationally manageable [32].

**Assumption 1.3** *The operator $F(x)$ satisfies Minty condition, if there exists a point $x^*$ such that*

$$\langle F(x), x - x^* \rangle \geq 0, \quad \text{for all } x \in \mathcal{X}. \tag{5}$$

The range of applications of nonmonotone VIs satisfying Minty conditions is quite extensive [20, 29, 39, 36, 66, 60]. We note, that this condition is weaker than monotonicity [30, 50, 59] and guarantees the existence of at least one strong solution since $F$ is continuous and $\mathcal{X}$ is closed and bounded [47]. To measure the optimality of point $\hat{x}$ we define the residue function $\text{RES}(\cdot) : \mathcal{X} \to \mathbb{R}_+$ [30, 50, 59]

$$\text{RES}(\hat{x}) = \sup_{x \in \mathcal{X}} \langle F(\hat{x}), \hat{x} - x \rangle \leq \varepsilon, \tag{6}$$

The boundedness of $\mathcal{X}$ and the existence of a strong solution ensure that the residual function is well-defined. If $\varepsilon = 0$, by (2), we get that $\hat{x}$ is a strong solution of VI.

**Contributions.** The main contribution of this paper lies in the development of a new second-order method robust to inexactness in the Jacobian, a common occurrence in machine learning. We demonstrate the algorithm's optimality in the monotone case by establishing a lower bound for this key setting. Expanding further:

1. We introduce a novel second-order algorithm, VIJI (Second-order Method for **V**ariational **I**nequalitues under **J**acoibian **I**nexactness), designed to handle $\delta$-inexact[3] Jacobian information. Specifically, in the context of smooth and monotone VIs, VIJI achieves a convergence rate of $O\left(\frac{\delta D^2}{T} + \frac{L_1 D^3}{T^{3/2}}\right)$ to find weak solution. For smooth nonmonotone VIs satisfying the Minty condition, we demonstrate a convergence rate of $O\left(\frac{\delta D^2}{\sqrt{T}} + \frac{L_1 D^3}{T}\right)$ to identify strong solution. Notably, when $\delta \leq \frac{L_1 D}{\sqrt{T}}$, our method matches the convergence rates of optimal exact second-order methods [2, 69].

2. We establish the optimal performance of our algorithm on monotone smooth operators by deriving a theoretical complexity lower bound of $\Omega\left(\frac{\delta D^2}{T} + \frac{L_1 D^3}{T^{3/2}}\right)$ to find weak solution for the case of $\delta$-inexact Jacobians.

3. Our algorithm involves solving a variational inequality subproblem. To tackle this challenge, we introduce an approximation condition, which makes the solution computationally feasible.

4. We introduce a new Quasi-Newton update for approximating the Jacobian, which significantly decreases the per-iteration cost of the algorithm while maintaining a global sublinear convergence rate. Numerical experiments demonstrate the practical benefits of our method.

5. We extend our algorithm for higher-order VIs with inexact high-order derivatives, resulting in $O\left(\sum_{i=1}^{p-1} \frac{\delta_i D^{i+1}}{T^{(i+1)/2}} + \frac{L_{p-1} D^{p+1}}{T^{(p+1)/2}}\right)$ rate for monotone and smooth VIs with $\delta_i$-inexact $i$-th derivative to find weak solution. Moreover, we extend our proposed high-order method to nonmonotone VIs.

6. We propose a restarted version of VIJI for strongly monotone VIs, which exhibits a linear rate.

**Comparison with Lin, Mertikopoulos, and Jordan [71].** To the best of our knowledge, the work [71] is the most closely related to our research. The objective of [71] was to develop a method for convex-concave unconstrained min-max optimization under inexact Jacobian information with an optimal convergence rate $O(\varepsilon^{-2/3})$ matching the lower bound $\Omega(\varepsilon^{-2/3})$ [69]. The authors successfully achieve this goal by proposing a second-order algorithm Inexact-Newton-MinMax

---

[3]A formal definition is provided in the following section.

method based on the Perseus [69]. To attain optimal convergence, they constrained the Jacobian inexactness with a function that decreases as the method converges and bounded the norm of Jacobian from above. While these assumptions might be suitable for randomized sampling in finite-sum and stochastic problems, they may not hold for many approximation strategies, such as Quasi-Newton algorithms. The aim of our work, however, is to study the impact of Jacobian inaccuracy on the convergence of second-order methods for VIs (a special case of which are min-max problems) and to identify the explicit dependence of the convergence rate on the inexactness. Compared to the work [71], the Jacobian inaccuracy directly affects the step sizes in our algorithm, allowing us to achieve a convergence rate of $O(\varepsilon^{-2/3} + \delta\varepsilon^{-1})$ for any given $\delta$. VIJI can be viewed as a generalization of Inexact-Newton-MinMax. With the same assumption on $\delta$ as in [71], our methods for min-max optimization are equivalent. The inexactness of the subproblem and the solution approach proposed in [71] remain valid for our method even with arbitrary large $\delta$. Further details about application of our method to min-max problems can be found in Appendix J.

## 2 Preliminaries

**Notation.** Let $\mathbb{R}^d$ be a finite-dimensional vector space with scalar product $\langle \cdot, \cdot \rangle$. For vector $x \in \mathbb{R}^d$ we denote Euclidean norm as $\|x\|$. For $X \in \mathbb{R}^{d_1 \times \cdots \times d_p}$, we define

$$X[z^1, \cdots, z^p] = \sum_{1 \leq i_j \leq d_j, 1 \leq j \leq p} (X_{i_1, \cdots, i_p}) z_{i_1}^1 \cdots z_{i_p}^p,$$

and $\|X\|_{\mathrm{op}} = \max_{\|z^i\|=1, 1 \leq j \leq p} X[z^1, \cdots, z^p]$. Fixing $p \geq 1$ and letting $F : \mathbb{R}^d \to \mathbb{R}^d$ be a continuous and high-order differentiable operator, we define $\nabla^{(p)} F(x)$ as the $p^{\mathrm{th}}$-order derivative at a point $x \in \mathbb{R}^d$. To be more precise, letting $z_1, \ldots, z_k \in \mathbb{R}^d$, we have

$$\nabla^{(k)} F(x)[z^1, \cdots, z^k] = \sum_{1 \leq i_1, \ldots, i_k \leq d} \left( \frac{\partial F_{i_1}}{\partial x_{i_2} \cdots \partial x_{i_k}}(x) \right) z_{i_1}^1 \cdots z_{i_k}^k.$$

**Taylor approximation and oracle feedback.** The starting point for our method is the first-order Taylor polynomial of the operator $F$ at point $v$ : $\Phi_v(x) = F(v) + \nabla F(v)[x - v]$. Since the computation of Jacobian $\nabla F(v)$ could be a quite tiresome task, it seems natural to introduce an inexact approximation $J(v)$. Based on it we introduce inexact Taylor approximation – one of the main building blocks of our algorithm

$$\Psi_v(x) = F(v) + J(v)[x - v], \quad v \in \mathbb{R}^d, \tag{7}$$

where $J(x)$ satisfies the following assumption.

**Assumption 2.1** *For given $v \in \mathcal{X}$ $\delta$-inexact Jacobian satisfies:*

$$\|\nabla F(v) - J(v)\| \leq \delta. \tag{8}$$

As it was shown, e.g. in [52], Assumption 1.2 allows to control the quality of approximation of operator $F$ by its Taylor polynomial

$$\|F(x) - \Phi_v(x)\| \leq \frac{L_1}{2} \|x - v\|^2, \quad x, v \in \mathcal{X}. \tag{9}$$

The next lemma is counterpart of (9) for the case of inexact Jacobian.

**Lemma 2.2** *Let Assumptions 1.2 and 2.1 hold. Then, for any $x, v \in \mathcal{X}$*

$$\|F(x) - \Psi_v(x)\| \leq \frac{L_1}{2} \|x - v\|^2 + \delta\|x - v\|.$$

## 3 VIJI algorithm

In this section, extending on recent optimal high-order method for MVIs Perseus [69], we present our proposed method, dubbed as VIJI and detailed in Algorithm 1.

**Algorithm 1** VIJI

---

**Input:** initial point $x_0 \in \mathcal{X}$, parameters $L_1$, $\eta$, sequence $\{\beta_k\}$, and $\mathsf{opt} \in \{0, 1, 2\}$.
**Initialization:** set $s_0 = 0 \in \mathbb{R}^d$.
**for** $k = 0, 1, 2, \ldots, T$ **do**
    Compute $v_{k+1} = \operatorname{argmax}_{v \in \mathcal{X}} \{\langle s_k, v - x_0 \rangle - \frac{1}{2}\|v - x_0\|^2\}$.
    Compute $x_{k+1} \in \mathcal{X}$ such that condition (12) holds true.
    Compute $\lambda_{k+1}$ such that $\frac{1}{32} \leq \lambda_{k+1} \left(\frac{L_1}{2}\|x_{k+1} - v_{k+1}\| + \beta_{k+1}\right) \leq \frac{1}{22}$.
    Compute $s_{k+1} = s_k - \lambda_{k+1} F(x_{k+1})$.
**Output:** $\hat{x} = \begin{cases} \tilde{x}_T = \frac{1}{\sum_{k=1}^{T} \lambda_k} \sum_{k=1}^{T} \lambda_k x_k, & \text{if } \mathsf{opt} = 0, \\ x_T, & \text{else if } \mathsf{opt} = 1, \\ x_{k_T} \text{ for } k_T = \operatorname{argmin}_{1 \leq k \leq T} \|x_k - v_k\|, & \text{else if } \mathsf{opt} = 2. \end{cases}$

---

**The model of objective and subproblem's solution.** We begin the description of the algorithm by introducing the inexact model of objective $\Omega_v^\eta(x)$

$$\Omega_v^\eta(x) = \Psi_v(x) + \eta\delta(x - v) + 5L_1\|x - v\|(x - v), \tag{10}$$

where $\eta > 0$ is given constant. Here, we introduced the additional regularization term $\eta\delta(x - v)$. As we will demonstrate, this term is crucial for ensuring that the method's subproblem has a solution. For brevity, we use a regularization constant of $5L$. Using larger coefficients would yield the same convergence rate. As other dual extrapolation-type methods, VIJI includes the following subproblem

$$\text{find } x_{k+1} \in \mathcal{X} \text{ such that } \langle \Omega_{v_{k+1}}^\eta(x_{k+1}), x - x_{k+1} \rangle \geq 0 \text{ for all } x \in \mathcal{X}. \tag{11}$$

First of all, the strong solution of this VI exists because $\Omega_{v_{k+1}}^\eta(x)$ is continuous, and $\mathcal{X}$ is a closed, bounded, and convex set. Next, we demonstrate that in a monotone setting VI (11) is also monotone.

**Lemma 3.1** *Let Assumptions* (1.1), (1.2), (2.1) *hold. Then for any* $x, v_{k+1} \in \mathcal{X}$ *VI* (11) *is monotone*

$$\frac{1}{2}\left(\nabla\Omega_v(x) + \nabla\Omega_v(x)^T\right) \succeq 4L_1\|x - v\|I_{d\times d} + 5L_1\frac{(x-v)(x-v)^T}{\|x-v\|} + (\eta - 1)\delta I_{d\times d}.$$

Following the work [69], one can find a strong solution to such VI using mirror-prox methods from [1], achieving the following approximate condition

$$\sup_{x \in \mathcal{X}} \langle \Omega_{v_{k+1}}(x_{k+1}), x_{k+1} - x \rangle \leq \frac{L_1}{2}\|x_{k+1} - v_{k+1}\|^3 + \delta\|x_{k+1} - v_{k+1}\|^2. \tag{12}$$

This ensures that the subproblem is computationally solvable in the monotone setting. In specific cases, such as minimax optimization, other efficient subsolvers can be employed [49, 2, 71].

**Adaptive dual stepsizes.** Adaptive stepsizes in dual space $\lambda_k$ are another core aspect of the algorithm. Due to inaccuracies in the Jacobian, applying the standard adaptive strategy, such as in the Perseus algorithm [69], can lead to excessively large steps, potentially slowing down the method. To address this issue, an additional term $\beta_k$ is incorporated into the adaptive strategy for selecting $\lambda_k$. Thus, when $\|x_k - v_k\|$ is small (indicating proximity to the optimum), $\beta_k$ has a greater influence on the choice of $\lambda_k$, preventing the method from taking overly aggressive steps. Similar behavior can be observed in accelerated second-order methods for minimization with inexact Hessians from theoretical [3, Lemma 5], [4, Appendix B, Lemma E.3] and practical [4, Section 7] perspectives.

**Convergence in monotone setting.** Now, we are prepared to present the convergence theorem of Algorithm 1 in the monotone case.

**Theorem 3.2** *Let Assumptions 1.1, 1.2, 2.1 hold. Then, after* $T \geq 1$ *iterations of Algorithm 1 with parameters* $\beta_k = \delta$, $\eta = 10$, $\mathsf{opt} = 0$, *we get the following bound*

$$\mathrm{GAP}(\tilde{x}_T) = \sup_{x \in \mathcal{X}} \langle F(x), \tilde{x}_T - x \rangle = O\left(\frac{L_1 D^3}{T^{3/2}} + \frac{\delta D^2}{T}\right). \tag{13}$$

The upper bound (13) consists of two terms. The first one corresponds to exact convergence and matches the lower bound for second-order VI methods [69]. The second term illustrates the impact

of the Jacobian's inexactness on the convergence rate, aligning with the lower bound for first-order methods [89]. We also notice that one can make the bound from (13) tighter for the so-called restricted gap-function [81] defined as $\text{RGAP}(y) = \sup_{x \in \mathcal{C}} \langle F(x), y - x \rangle$, where $\mathcal{C} \subseteq \mathcal{X}$ and the solution set $\mathcal{X}^*$ of (2) satisfies $\mathcal{X}^* \subseteq \mathcal{C}$. In particular, following similar steps as in the proof of Theorem 3.2, one can derive $O\left(\frac{L_1 \widehat{D}^3}{T^{3/2}} + \frac{\delta \widehat{D}^2}{T}\right)$ bound for $\text{RGAP}(\tilde{x}_T)$, where $\widehat{D} = \sup_{x \in \mathcal{C}} \|x - x_0\|$.

In the next theorem, we show that the method can achieve the optimal convergence rate of second-order methods under additional assumption on $\delta$.

**Theorem 3.3** *Let Assumptions 1.1, 1.2, 2.1 hold. Let $\{x_k, v_k\}$ be iterates generated by Algorithm 1 and*

$$\|(\nabla F(v_k) - J(v_k))[x_k - v_k]\| \leq \delta_k \|x_k - v_k\|, \quad \delta_k \leq \frac{L_1}{2}\|x_k - v_k\|. \tag{14}$$

*Then, after $T \geq 1$ iterations of Algorithm 1 with parameters $\beta_k = \frac{L_1}{2}\|x_k - v_k\|$, $\eta = 10$, $\textbf{opt} = 0$, we get the following bound*

$$\text{GAP}(\tilde{x}_T) = \sup_{x \in \mathcal{X}} \langle F(x), \tilde{x}_T - x \rangle \leq O\left(\frac{L_1 D^3}{T^{3/2}}\right).$$

Note, that condition (14) is verifiable, indicating that the method can adjust to $\delta_k$ in cases of controllable inexactness. Specifically, at each iteration, we can solve subproblem (12). If the assumption regarding $\delta_k$ is not met, we can improve Jacobian approximation and repeat procedure. Moreover, similarly to the previous theorem, one can tighten the above bound for $\text{RGAP}(\tilde{x}_T)$ and get the dependence on $\widehat{D} = \sup_{x \in \mathcal{C}} \|x - x_0\|$ instead of $D$.

**Convergence in nonmonotone setting.** To begin with, in the nonmonotone case, the subproblem (11) may not exhibit monotonicity, and solving (12) becomes challenging [32]. Yet, in certain specific scenarios, such as unconstrained minimization tasks, it remains feasible to find a solution by leveraging the cubic structure of the subproblem [26]. The following theorem establishes the convergence of VIJI in the nonmonotone setting.

**Theorem 3.4** *Let Assumptions 1.2, 1.3, 2.1 hold. Then after $T \geq 1$ iterations of Algorithm 1 with parameters $\beta_k = \delta, \eta = 10, \textbf{opt} = 2$ we get the following bound*

$$\text{RES}(\hat{x}) = \sup_{x \in \mathcal{X}} \langle F(\hat{x}_T), \hat{x}_T - x \rangle = O\left(\frac{L_1 D^3}{T} + \frac{\delta D^2}{\sqrt{T}}\right).$$

The convergence rate could be improved by additional assumption on inexact Jacobian.

**Theorem 3.5** *Let Assumptions 1.2, 1.3, 2.1 hold. Let $\{x_k, v_k\}$ be iterates generated by Algorithm 1 that satisfy (14). Then after $T \geq 1$ iterations of Algorithm 1 with parameters $\beta_k = \frac{L}{2}\|x_k - v_k\|$, $\eta = 10, \textbf{opt} = 2$ we get the following bound*

$$\text{RES}(\hat{x}) = \sup_{x \in \mathcal{X}} \langle F(\hat{x}_T), \hat{x}_T - x \rangle = O\left(\frac{L_1 D^3}{T}\right).$$

# 4 The lower bound

In this section, we establish a theoretical lower bound for the complexity of first-order algorithms using inexact Jacobians for monotone MVIs. The proof technique draws inspiration from works [33, 4] and based on lower bounds from [69, 89].

We start by describing the available information and the method's structure. The considered class of algorithms relies on data provided by a first-order $\delta$-inexact oracle, denoted as $\mathcal{O} : \mathcal{X} \to \mathbb{R}^d \times \mathbb{R}^{d \times d}$. Given a point $\bar{x} \in \mathcal{X}$, the oracle returns

$$\mathcal{O}(\bar{x}) = (F(\bar{x}), J(\bar{x})), \text{ such that Assumption 2.1 holds.} \tag{15}$$

The method is able to generate points $\{x_k\}_{k \geq 0}$ that satisfy the following condition

$$s \in \text{Lin}(F(x_0), \ldots, F(x_k)), \qquad \bar{x} = \text{argmax}_{x \in \mathcal{X}} \{\langle s, x - x_0 \rangle - \frac{1}{2}\|x - x_0\|^2\},$$
$$x_{k+1} \in \mathcal{X} \text{ satisfies that } \langle \Omega_{\bar{x}}(x_{k+1} - x_k), x - x_{k+1} \rangle \geq 0 \text{ for all } x \in \mathcal{X},$$

where $\Omega_{\bar{x}}(h) = a_1 F(\bar{x}) + a_2 J(\bar{x})[h] + b_1 h + b_2 \|h\| h$.

Next, we state the primary assumption concerning the method's ability to generate new points.

**Assumption 4.1** *The method generates a recursive sequence of iterates $\{x_k\}_{k \geq 0}$ that satisfies the following condition: for all $k \geq 0$, we have that $x_{k+1} \in \mathcal{X}$ satisfies that $\langle \Omega_{\bar{x}}(x_{k+1} - x_k), x - x_{k+1} \rangle \geq 0$ for all $x \in \mathcal{X}$, where*

$$\bar{x} = \operatorname*{argmax}_{x \in \mathcal{X}}\{\langle s, x - x_0 \rangle - \tfrac{1}{2}\|x - x_0\|^2\} \text{ and } s \in \operatorname{Lin}(F(x_0), \dots, F(x_k)).$$

As highlighted in [69], Assumption 4.1 is suitably satisfied by various dual extrapolation methods. However, it might not be applicable to alternative methods for variational inequalities, such as extragradient methods and their variants. We leave the generalization of inexact lower bounds for these algorithms to future research, as even lower bounds for exact algorithms [2, 69] do not address this case. Now, let us introduce the generalization of smoothness

**Assumption 4.2** *The operator $F(x)$ is $i$-th-order $L_i$-smooth ($i \geq 0$), if it has Lipschitz-continuous $i$-th-order derivative*

$$\|\nabla^i F(x) - \nabla^i F(y)\|_{\mathrm{op}} \leq L_i \|x - y\|, \quad \text{for all } x, y \in \mathcal{X}. \tag{16}$$

Finally, we present the lower bound theorem for first-order methods with inexact Jacobians.

**Theorem 4.3** *Let some first-order method $\mathcal{M}$ satisfy Assumption 4.1 and have access only $\delta$-inexact first-order oracle 15. Assume the method $\mathcal{M}$ ensures for any $L_0$-zero-order smooth and $L_1$-first-order smooth monotone operator $F$ the following convergence rate*

$$\operatorname{GAP}(\hat{x}) \leq O(1) \max\left\{\frac{\delta D^2}{\Xi_1(T)}; \frac{L_1 D^3}{\Xi_2(T)}\right\}. \tag{17}$$

*Then for all $T \geq 1$ we have*

$$\Xi_1(T) \leq T, \qquad \Xi_2(T) \leq T^{3/2}. \tag{18}$$

## 5 Quasi-Newton Approximation

In this section, inspired by Quasi-Newton (QN) methods for Hessian approximation, we propose QN approximations for Jacobians. Our goal is to create a simple scheme to approximate the first-order derivative and thereby reduce the complexity of the subproblem. We compute $J_x$ using a QN update and use it as an inexact Jacobian in the model $\Omega_v^\eta(x)$ (10).

$$J_x = J^r = J^0 + \sum_{i=0}^{r-1} c_i u_i v_i^\top = J^0 + U^\top C V, \tag{19}$$

where $r$ is a rank of approximation, $J^0 \in \mathbb{R}^{d \times d} \succeq 0$, $u_i \in \mathbb{R}^d$, $v_i \in \mathbb{R}^d$ are known. $U \in \mathbb{R}^{r \times d}$ and $V \in \mathbb{R}^{r \times d}$ are matrices of stacked vectors $U = [u_0, \dots, u_{r-1}]$ and $V = [v_0, \dots, v_{r-1}]$ and $C \in \mathbb{R}^{r \times r}$ is a diagonal matrix $C = \operatorname{diag}([c_0, \dots, c_{r-1}])$. If $u_i = v_i$ the update becomes symmetric.

**L-Broyden** is a non-symmetric variant of QN approximation [45, 38] of the following form

$$J^{i+1} = J^i + \frac{(y_i - J^i s_i)s_i^\top}{s_i^\top s_i}, \quad \forall i = 0, \dots, m-1. \tag{20}$$

In a view of (19), this update is obtained by setting $u_i = y_i - J^i s_i$, $v_i = s_i$, $c_i = 1/(s_i^\top s_i)$, and the rank $r = m$ is equal to memory size $m$.

***Damped* L-Broyden** is another option for QN approximation with non-symmetric damped update

$$J^{i+1} = J^i + \frac{1}{m+1}\frac{(y_i - J^i s_i)s_i^\top}{s_i^\top s_i}, \quad \forall i = 0, \dots, m-1. \tag{21}$$

By choosing $u_i = y_i - J^i s_i$, $v_i = s_i$, $c_i = 1/((m+1)s_i^\top s_i)$, $r = m$, we derive this update from (19). We define the matrix $J^r(J^0, U, V, C) = J^r(J^0, Y, S, C)$, where $Y$ and $S$ are formed by stacking the vectors $[y_0, \dots, y_{m-1}]$ and $[s_0, \dots, s_{m-1}]$. The matrix $J^m(J^0, Y, S)$ can be computed for any given pair $(Y, S)$. Next, we describe two strategies for the choice of $(s, y)$ pairs used in (20), (21).

**QN with operator history** is the well-known classic variant where operator differences are stored:

$$s_i = z_{i+1} - z_i, \qquad y_i = F(z_{i+1}) - F(z_i).$$

This approach is computationally efficient as it does not require additional operator calculations.

**QN with JVP sampling** is based on fast computation of Jacobian-Vector Products (JVP):

$$y_i = \nabla F(x)s_i,$$

where $s_i$ are random vectors uniformly distributed on the unit sphere such that $\|s_i\| = 1$ and $s_0, \ldots, s_{m-1}$ are linearly independent. Note, for $m \ll d$, each $s_i$ is linearly independent with high probability. This approach requires only $m$ operator/JVP computations per step, which is considerably fewer than the $d$ JVPs needed for a full Jacobian. Utilizing the current Jacobian information allows to improve the accuracy of the approximation.

In the following theorem, we demonstrate that these approximations satisfy Assumption 2.1 and condition (8) for both the QN with operator history and JVP sampling methods.

**Theorem 5.1** *Let $F(x)$ be $L_0$-zero-order smooth operator. For $m$-memory L-Broyden approximation of the Jacobian $J_x = J^m$ defined iteratively by (20) with $0 \preceq J^0 \preceq L_0 I$, we have $\delta \leq (m+2)L_0$. For $m$-memory Damped L-Broyden approximation $J_x = J^m$ of the Jacobian defined iteratively by (21) with $0 \preceq J^0 \preceq \frac{L_0}{m+1} I$, the condition $\delta \leq 2L_0$ holds true.*

With the primary toolkit for QN approximation in VIs established, we can now discuss efficient way of solving the subproblem (11), which takes the following form

$$\text{find } y \in \mathcal{X} \text{ such that } \langle F(x) + (J_x + \eta\delta I + 5L_1\|y-x\|I)(y-x), z-y \rangle \geq 0 \text{ for all } z \in \mathcal{X}. \quad (22)$$

Let us introduce a parameter $\tau = \|y - x\|$ for a segment search in $\tau \in [0; D]$. To solve (22), we consider another problem

$$\text{find } y_\tau \in \mathcal{X} \text{ such that } \langle A_\tau^{-1}F(x) + y_\tau - x, z - y_\tau \rangle \geq 0 \text{ for all } z \in \mathcal{X}, \quad (23)$$

where $A_\tau = J_x + (\eta\delta + 5L_1\tau)I$. Problems (22) and (23) are equivalent when $\tau = \|y_\tau - x\|$. The subproblem (23) can be reformulated as minimization problem

$$y_\tau = \underset{y \in \mathcal{X}}{\operatorname{argmin}} \left\{ \langle A_\tau^{-1}F(x), y - x \rangle + \tfrac{1}{2}\|y - x\|^2 \right\}, \quad (24)$$

where (23) is an optimality condition for (24). The goal is to find $y_\tau$ such that $\upsilon(\tau) = |\tau - \|y_\tau - x\|| \leq \varepsilon$. As $\upsilon(\tau)$ is a continuous function of $\tau$, we can find this solution via bisection segment-search with $\log_2 \frac{D}{\varepsilon}$ iterations. This ray-search procedure is similar to the subproblem solution for the Cubic Regularized Newton subproblem.

For $r$-rank QN approximation $J^r$ from (19), we can effectively compute $A_\tau^{-1}F(x)$, where $A_\tau = J^r + (\eta\delta + 5L_1\tau)I = U^\top CV + J^0 + (\eta\delta + 5L_1\tau)I = U^\top CV + B$ and $B = J^0 + (\eta\delta + 5L_1\tau)I$ by using the Woodbury matrix identity[96, 97].

$$A_\tau^{-1}F(x) = \left(B + U^\top CV\right)^{-1} F(x) = B^{-1}F(x) - B^{-1}U^\top(C^{-1} + VB^{-1}U^\top)^{-1}VB^{-1}F(x).$$

For computational efficiency, it is better to choose $J^0$ as a diagonal matrix, then inversion $B^{-1}$ is computed by $O(d)$ arithmetical operations, $C^{-1}$ by $O(r)$ operations. $VB^{-1}U^\top$ requires $O(r^2d)$ for classical multiplication and can be improved by fast matrix multiplication. $(C^{-1} + VB^{-1}U^\top)^{-1}$ can be computed by $O(r^3)$, as an inverse of $r$-rank matrix. The rest of the operations are cheaper. Thus, the total number of arithmetic operations is $O(r^2d)$ instead of $O(d^3)$ for Jacobian inversion. We need to perform this inversion logarithmic number of times. Therefore, the total computational cost with the segment-search procedure is $\tilde{O}\left(r^2d + r^3\log_2(\frac{D}{\varepsilon})\right)$.

## 6 Strongly monotone setting

**Assumption 6.1** *The operator $F : \mathbb{R}^d \to \mathbb{R}^d$ is called strongly monotone if there exists a constant $\mu > 0$ such that*

$$\langle F(x) - F(y), x - y \rangle \geq \mu\|x - y\|^2, \quad \text{for all } x, y \in \mathcal{X}. \quad (25)$$

To leverage the strong monotonicity of the objective function and achieve a linear convergence rate, we introduce the restarted version of Algorithm 1 dubbed as VIJI-Restarted. Restart techniques are widely utilized in optimization and typically preserve optimality. In other words, restarting an optimal method for convex functions or monotone problems effectively transforms it into an optimal method for strongly convex or strongly monotone problems. Within iteration of VIJI-Restarted listed as Algorithm 2, we execute VIJI for a predefined number of iterations (26). Next, the output of this run is used as initial point for next run of VIJI with parameters reset, and this iterative process continues.

**Algorithm 2** VIJI-Restarted

---

**Input:** initial point $z_0 \in \mathcal{X}$, $D = \max_{x,y \in \mathcal{X}}$, parameters $L, \delta$.
**Initialization**: $n = \lceil \log \frac{D}{\varepsilon} \rceil$, $R = D$.
**for** i=1, ..., n **do**
    Set $x_0 = z_{i-1}$, $R_{i-1} = \frac{R}{2^{i-1}}$.
    Run Algorithm 1 with $\beta_k = \delta$, $\eta = 10$, $\mathsf{opt} = 0$ for $T_i$ iterations, where

$$T_i = O(1) \left\lceil \max \left\{ \frac{L_1^{2/3} R_{i-1}^{2/3}}{\mu^{2/3}}, \frac{\delta}{\mu} \right\} \right\rceil. \tag{26}$$

    Set $z_i = x_{T_i}$.

---

**Theorem 6.2** *Let Assumptions 1.2, 2.1, 6.1 hold. Then the total number of iterations of Algorithm 2 to reach desired accuracy $\|z_s - x^*\| \leq \varepsilon$, where $x^*$ is the solution of (2) is*

$$O\left( \left( \frac{L_1 D}{\mu} \right)^{\frac{2}{3}} + \left( \frac{\delta}{\mu} + 1 \right) \left\lceil \log \frac{D}{\varepsilon} \right\rceil \right).$$

## 7 Tensor generalization

In this section, we generalize the results presented in Section 3 to the $p$-th order case. We consider a higher-order method with inexact high-order derivatives, satisfying the following assumption.

**Assumption 7.1** *For all $x, v \in \mathcal{X}$, $i \geq 1$, $i$-th inexact derivative of $F$, which we denote as $G_i$, satisfies*

$$\left\| \left( \nabla^i F(v) - G_i(v) \right) [x - v]^{i-1} \right\| \leq \delta_i \|x - v\|^{i-1}. \tag{27}$$

Based on inexact $(p-1)$-th-order Taylor approximation $\Psi_{p,v}(x) = F(v) + \sum_{i=1}^{p-1} \frac{1}{i!} \nabla^i G_i(v)[x-v]^i$, we introduce the inexact tensor model $\Omega_{p,v}(x)$ of the objective

$$\Omega_{p,v}(x) = \Psi_{p,v}(x) + \sum_{i=1}^{p-1} \frac{\eta_i \delta_i}{i!} \|x - v\|^{i-1}(x - v) + \frac{5L_{p-1}}{(p-1)!} \|x - v\|^{p-1}(x - v). \tag{28}$$

The tensor generalization of Algorithm 1, referred to as VIHI (High-order Method for **V**ariational **I**nequalites under **H**igh-order derivatives **I**nexactness ) and detailed in Appendix G, involves the inexact solution of the subproblem, which satisfies the following condition:

$$\sup_{x \in \mathcal{X}} \langle \Omega_{v_{k+1}}(x_{k+1}), x_{k+1} - x \rangle \leq \frac{L_{p-1}}{p!} \|x_{k+1} - v_{k+1}\|^{p+1} + \sum_{i=1}^{p-1} \frac{\delta_i}{i!} \|x_{k+1} - v_{k+1}\|^{i+1}.$$

Another difference in VIHI compared to VIJI (Algorithm 1) is the adaptive strategy for $\lambda_{k+1}$:

$$\frac{1}{4(5p-2)} \leq \lambda_k \left( \frac{L_{p-1}}{p!} \|x_{k+1} - v_{k+1}\|^{p+1} + \sum_{i=1}^{p-1} \frac{\delta_i}{i!} \|x_{k+1} - v_{k+1}\|^{i+1} \right) \leq \frac{1}{2(5p+1)}.$$

The other steps of Algorithm 1 remain unchanged for the higher-order method. Now, we are ready to present the convergence properties of VIHI.

**Theorem 7.2** *Let Assumptions 1.1, 4.2 with $i = p - 1$, and 7.1 hold. Then, after $T \geq 1$ iterations of VIHI with parameters $\eta_i = 5p$, $\mathsf{opt} = 0$, we get the following bound*

$$\mathrm{GAP}(\tilde{x}_T) = \sup_{x \in \mathcal{X}} \langle F(x), \tilde{x}_T - x \rangle \leq O\left( \frac{L_{p-1} D^{p+1}}{T^{\frac{p+1}{2}}} + \sum_{i=1}^{p-1} \frac{\delta_i D^{i+1}}{T^{\frac{i+1}{2}}} \right).$$

Finally, we extend our tensor generalization to nonmonotone case and obtain the following result.

**Theorem 7.3** *Let Assumptions 1.3, 4.2 with $i = p - 1$, and 7.1 hold. Then after $T \geq 1$ iterations of VIHI with parameters $\eta = 5p$, $\mathsf{opt} = 2$ we get the following bound*

$$\mathrm{RES}(\hat{x}) := \sup_{x \in \mathcal{X}} \langle F(\hat{x}_t), \hat{x}_t - x \rangle = O\left( \frac{L_{p-1} D^{p+1}}{T^{\frac{p}{2}}} + \sum_{i=1}^{p-1} \frac{\delta_i D^{i+1}}{T^{\frac{i}{2}}} \right).$$

# 8 Experiments

In this section, we present numerical experiments to demonstrate the efficiency of our proposed methods. We consider the cubic regularized bilinear min-max problem of the form:

$$\min_{x\in\mathbb{R}^d}\max_{y\in\mathbb{R}^d} f(x,y) = y^\top(Ax - b) + \frac{\rho}{6}\|x\|^3,$$

where $\rho > 0$, $b = [1,0,\ldots,0] \in \mathbb{R}^d$, and $A \in \mathbb{R}^{d\times d}$, with all $1$ on the main diagonal and all $-1$ on the upper diagonal, the rest elements are $0$. To reformulate it as variational inequality, we define $F(x) = [\nabla_x f(x,y), -\nabla_y f(x,y)]$. This problem is inspired by the first-order lower bound function for variational inequalities and min-max problems and is commonly used to verify the convergence of high-order methods for VI [71, 52].

We implement our second-order method for **V**ariational **I**nequalities with **Q**uasi-Newton **A**pproximation (VIQA) as a PyTorch optimizer. The code is available in the OPTAMI package[4] [58]. VIQA Broyden refers to L-Broyden approximation (20) and VIQA Damped Broyden to (21), which are used as inexact Jacobians in VIJI (Algorithm 1). We compare them with the Extragradient method (EG) [62], first-order Perseus (Perseus1), and second-order Perseus with Jacobian (Perseus2).

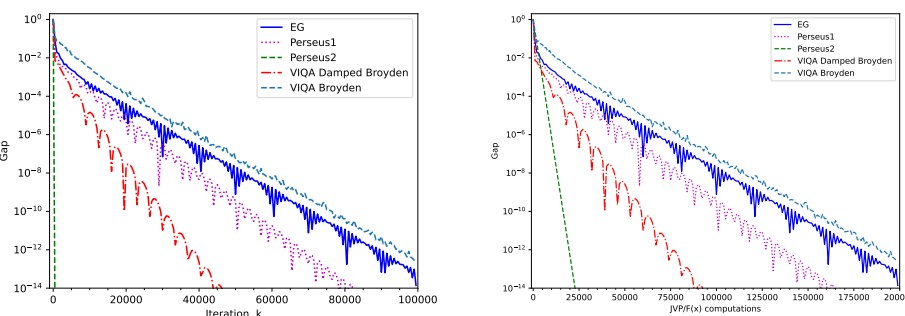

Figure 1: Comparison of different methods for $d = 50, \rho = 1e - 3$.

In Figure 1, one can see that second-order information in Perseus2 significantly accelerates the convergence compared to Perseus1. However, it is expensive to compute Jacobian and solve second-order subproblems every iteration. VIQA with the proposed Damped Broyden approximation (21) is significantly faster than EG, Perseus1, and VIQA Broyden (20). It shows that this approximation improves the convergence of first-order Perseus1 and confirms the theoretical result that Damped Broyden is a more accurate approximation than classical Broyden from Theorem 5.1. The detailed parameters and setup are presented in Appendix I.

# 9 Conclusion

In this work, we introduced a second-order method specifically designed to handle Jacobian inexactness. We demonstrated its optimality in the monotone case by introducing a new lower bound and extended its applicability to tensor methods. However, similar to other high-order methods with global convergence properties, our algorithm involves a subproblem that necessitates an additional subroutine for its solution. To address this challenge, we proposed a computationally feasible criterion for solving the subproblem and implemented Quasi-Newton approximations for Jacobians, resulting in a significant reduction in per-iteration cost. Future investigations could explore incorporating inexactness within the operator itself and developing adaptive schemes to dynamically adjust for the level of inexactness encountered during the optimization process. Another open problem is a design of more accurate Quasi-Newton approximations specifically for Jacobians, focusing on non-symmetrical structure of Jacobian and specific inexactness criteria such as Assumption 2.1.

---

[4]`https://github.com/OPTAMI/OPTAMI`

**Acknowledgments.**

The work was supported by MIPT based Center of National Technology Initiatives in the field of Artificial Intelligence for the purposes of the "road map" of Artificial Intelligence development up to 2030 and supported by NTI Foundation (agreement No.70-2021-00207 dated 22.11.2021, identifier 000000S507521QYL0002).

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

# Contents

# A  Proofs of Lemmas 2.2, 3.1

In this section, we let $L := L_1$.

**Lemma 2.2** *Let Assumptions 1.2 and 2.1 hold. Then, for any $x, v \in \mathcal{X}$*

$$\|F(x) - \Psi_v(x)\| \le \tfrac{L}{2}\|x - v\|^2 + \delta\|x - v\|. \tag{29}$$

*Proof.* For any $x, y \in \mathcal{X}$

$$\|F(x) - \Psi_v(x)\| \le \|F(x) - \Phi_v(x)\| + \|\Phi_v(x) - \Psi_v(x)\|$$

$$\overset{(9)}{\le} \frac{L}{2}\|x - v\|^2 + \|(\nabla F(v) - J(v))[x - v]\| \le \frac{L}{2}\|x - v\|^2 + \delta\|x - v\|.$$

$\square$

**Lemma 3.1** *Let Assumptions* (1.1), (1.2), (2.1) *hold. Then for any* $x, v \in \mathcal{X}$ *VI* (11) *is monotone*

$$\tfrac{1}{2}\left(\nabla\Omega_v(x) + \nabla\Omega_v(x)^T\right) \succeq 4L_1\|x - v\|I_{d\times d} + 5L_1\frac{(x-v)(x-v)^T}{\|x-v\|} + (\eta - 1)\delta I_{d\times d}.$$

*Proof.* For all $x,\ v \in \mathcal{X}$

$$\tfrac{1}{2}\left(\nabla\Omega_v(x) + \nabla\Omega_v(x)^T\right)$$

$$\overset{(10)}{=} \tfrac{1}{2}\left(J(v) + J(v)^T\right) + \eta\delta I_{d\times d} + 5L\|x - v\|I_{d\times d} + 5L\frac{(x-v)(x-v)^T}{\|x-v\|}$$

$$\overset{(29)}{\succeq} \tfrac{1}{2}\left(\nabla F(x) + \nabla F(x)^T\right) + 4L\|x - v\|I_{d\times d} + 5L\frac{(x-v)(x-v)^T}{\|x-v\|} + (\eta - 1)\delta I_{d\times d}$$

$$\overset{(3)}{\succeq} 4L\|x - v\|I_{d\times d} + 5L\frac{(x-v)(x-v)^T}{\|x-v\|} + (\eta - 1)\delta I_{d\times d}.$$

$\square$

# B   Proofs of Theorems 3.2, 3.3

## B.1   Proof of Theorem 3.2

In this section, we let $L := L_1$. To show the convergence of Algorithm 1, we define the following Lyapunov function

$$\mathcal{E}_k = \max_{v\in\mathcal{X}} \langle s_k, v - x_0\rangle - \tfrac{1}{2}\|v - x_0\|^2. \tag{30}$$

**Lemma B.1** *Let Assumption 1.2, 2.1 hold. Then, for every integer $T \geq 1$, we have*

$$\sum_{k=1}^{T} \lambda_k\langle F(x_k), x_k - x\rangle \leq \mathcal{E}_0 - \mathcal{E}_T + \langle s_T, x - x_0\rangle - \tfrac{1}{8}\left(\sum_{k=1}^{T}\|x_k - v_k\|^2\right), \quad \text{for all } x \in \mathcal{X}.$$

*Proof.* By the definition of Lyapunov function (30) and Step 2 of Algorithm 1, we have

$$\mathcal{E}_k = \langle s_k, v_{k+1} - x_0\rangle - \tfrac{1}{2}\|v_{k+1} - x_0\|^2.$$

Then, we have

$$\mathcal{E}_{k+1} - \mathcal{E}_k = \langle s_{k+1}, v_{k+2} - x_0\rangle - \langle s_k, v_{k+1} - x_0\rangle - \tfrac{1}{2}\left(\|v_{k+2} - x_0\|^2 - \|v_{k+1} - x_0\|^2\right)$$
$$= \langle s_{k+1} - s_k, v_{k+1} - x_0\rangle + \langle s_{k+1}, v_{k+2} - v_{k+1}\rangle - \tfrac{1}{2}\left(\|v_{k+2} - x_0\|^2 - \|v_{k+1} - x_0\|^2\right). \tag{31}$$

By the update formula for $v_{k+1}$, we get

$$\langle x - v_{k+1}, s_k - v_{k+1} + x_0\rangle \leq 0, \quad \text{for all } x \in \mathcal{X}.$$

Letting $x = v_{k+2}$ in this inequality and using $\langle a, b\rangle = \tfrac{1}{2}(\|a + b\|^2 - \|a\|^2 - \|b\|^2)$, we have

$$\langle s_k, v_{k+2} - v_{k+1}\rangle \leq \langle v_{k+1} - x_0, v_{k+2} - v_{k+1}\rangle = \tfrac{1}{2}\left(\|v_{k+2} - x_0\|^2 - \|v_{k+1} - x_0\|^2 - \|v_{k+2} - v_{k+1}\|^2\right). \tag{32}$$

Plugging Eq. (32) into Eq. (31) and using Step 5 of Algorithm 1, we obtain:

$$\mathcal{E}_{k+1} - \mathcal{E}_k \overset{(32)}{\leq} \langle s_{k+1} - s_k, v_{k+1} - x_0\rangle + \langle s_{k+1} - s_k, v_{k+2} - v_{k+1}\rangle - \tfrac{1}{2}\|v_{k+2} - v_{k+1}\|^2$$

$$= \langle s_{k+1} - s_k, v_{k+2} - x_0\rangle - \tfrac{1}{2}\|v_{k+2} - v_{k+1}\|^2 \leq \lambda_{k+1}\langle F(x_{k+1}), x_0 - v_{k+2}\rangle - \tfrac{1}{2}\|v_{k+2} - v_{k+1}\|^2$$

$$= \lambda_{k+1}\langle F(x_{k+1}), x_0 - x\rangle + \lambda_{k+1}\langle F(x_{k+1}), x - x_{k+1}\rangle + \lambda_{k+1}\langle F(x_{k+1}), x_{k+1} - v_{k+2}\rangle - \tfrac{1}{2}\|v_{k+2} - v_{k+1}\|^2,$$

for any $x \in \mathcal{X}$. Summing up this inequality over $k = 0, 1, \ldots, T - 1$ and changing the counter $k + 1$ to $k$ yields that

$$\sum_{k=1}^{T} \lambda_k\langle F(x_k), x_k - x\rangle \leq \mathcal{E}_0 - \mathcal{E}_T + \underbrace{\sum_{k=1}^{T} \lambda_k\langle F(x_k), x_0 - x\rangle}_{\text{I}} + \underbrace{\sum_{k=1}^{T} \lambda_k\langle F(x_k), x_k - v_{k+1}\rangle - \tfrac{1}{2}\|v_k - v_{k+1}\|^2}_{\text{II}}.$$

$$\tag{33}$$

Using the update formula for $s_{k+1}$ and letting $s_0 = 0_d \in \mathbb{R}^d$, we have

$$\mathbf{I} = \sum_{k=1}^{T} \langle \lambda_k F(x_k), x_0 - x \rangle = \sum_{k=1}^{T} \langle s_{k-1} - s_k, x_0 - x \rangle = \langle s_0 - s_T, x_0 - x \rangle = \langle s_T, x - x_0 \rangle. \quad (34)$$

Since $x_{k+1} \in \mathcal{X}$ satisfies (12), we have

$$\langle \Omega_{v_k}^{\eta}(x_k), x - x_k \rangle \geq -\tfrac{L}{2}\|x_k - v_k\|^3 - \delta\|x_k - v_k\|^2, \quad \text{for all } x \in \mathcal{X}, \quad (35)$$

where $\Omega_v^{\eta}(x) : \mathbb{R}^d \to \mathbb{R}^d$ is defined in (10). Letting $x = v_{k+1}$ in (35), we have

$$\langle \Omega_{v_k}^{\eta}(x_k), x_k - v_{k+1} \rangle \leq \tfrac{L}{2}\|x_k - v_k\|^3 + \delta\|x_k - v_k\|^2. \quad (36)$$

Then,

$$\langle F(x_k), x_k - v_{k+1} \rangle$$
$$= \langle F(x_k) - \Omega_{v_k}^{\eta}(x_k) + \eta\delta(x_k - v_k) + 5L\|x_k - v_k\|(x_k - v_k), x_k - v_{k+1} \rangle$$
$$+ \langle \Omega_{v_k}^{\eta}(x_k), x_k - v_{k+1} \rangle - 5L\|x_k - v_k\|\langle x_k - v_k, x_k - v_{k+1} \rangle - \eta\delta\langle x_k - v_k, x_k - v_{k+1} \rangle$$

$$\overset{\text{Lem. (2.2), (36)}}{\leq} \tfrac{L}{2}\|x_k - v_k\|^2\|x_k - v_{k+1}\| + \delta\|x_k - v_k\|\|x_k - v_{k+1}\| + \tfrac{L}{2}\|x_k - v_k\|^3 + \delta\|x_k - v_k\|^2$$
$$-5L\|x_k - v_k\|\langle x_k - v_k, x_k - v_{k+1} \rangle - \eta\delta\langle x_k - v_k, x_k - v_{k+1} \rangle$$

Next, using $\langle x_k - v_k, x_k - v_{k+1} \rangle \geq \|x_k - v_k\|^2 - \|x_k - v_k\|\|v_k - v_{k+1}\|$ and $\|x_k - v_{k+1}\| \leq \|x_k - v_k\| + \|v_k - v_{k+1}\|$, we get

$$\langle F(x_k), x_k - v_{k+1} \rangle$$
$$\leq \tfrac{L}{2}\|x_k - v_k\|^3 + \tfrac{L}{2}\|x_k - v_k\|^2\|v_k - v_{k+1}\| + \delta\|x_k - v_k\|^2 + \delta\|x_k - v_k\|\|v_k - v_{k+1}\|$$
$$+ \tfrac{L}{2}\|x_k - v_k\|^3 + \delta\|x_k - v_k\|^2 - 5L\|x_k - v_k\|^3 + 5L\|x_k - v_k\|^2\|v_k - v_{k+1}\| - \eta\delta\|x_k - v_k\|^2$$
$$+ \eta\delta\|x_k - v_k\|\|v_k - v_{k+1}\|$$
$$= \tfrac{11L}{2}\|x_k - v_k\|^2\|v_k - v_{k+1}\| - 4L\|x_k - v_k\|^3 + (\eta + 1)\delta\|x_k - v_k\|\|v_k - v_{k+1}\| - (\eta - 2)\delta\|x_k - v_k\|^2$$

Next,

$$\mathbf{II} \leq \sum_{k=1}^{T} \left( \tfrac{11\lambda_k L}{2}\|x_k - v_k\|^2\|v_k - v_{k+1}\| - 4\lambda_k L\|x_k - v_k\|^3 \right.$$
$$+ (\eta + 1)\delta\lambda_k\|x_k - v_k\|\|v_k - v_{k+1}\| - (\eta - 2)\delta\lambda_k\|x_k - v_k\|^2 - \tfrac{1}{2}\|v_k - v_{k+1}\|^2 \Big)$$
$$\leq \sum_{k=1}^{T} \left( \tfrac{1}{2}\|x_k - v_k\|\|v_k - v_{k+1}\| - \tfrac{1}{4}\|x_k - v_k\|^2 - \tfrac{1}{2}\|v_k - v_{k+1}\|^2 \right),$$

where the last inequality is due to the following choice of $\eta = 10$ and $\lambda$ : $\tfrac{1}{32} \leq \lambda_k \left( \tfrac{L}{2}\|x_k - v_k\| + \delta \right) \leq \tfrac{1}{22}$. Then,

$$\mathbf{II} \leq \sum_{k=1}^{T} \left( \tfrac{1}{2}\|x_k - v_k\|\|v_k - v_{k+1}\| - \tfrac{1}{4}\|x_k - v_k\|^2 - \tfrac{1}{2}\|v_k - v_{k+1}\|^2 \right)$$
$$\leq -\tfrac{1}{8}\left( \sum_{k=1}^{T}\|x_k - v_k\|^2 \right). \quad (37)$$

Plugging (34) and (37) into (33) yields that

$$\sum_{k=1}^{T} \lambda_k \langle F(x_k), x_k - x \rangle \leq \mathcal{E}_0 - \mathcal{E}_T + \langle s_T, x - x_0 \rangle - \tfrac{1}{8}\left( \sum_{k=1}^{T}\|x_k - v_k\|^2 \right).$$

$\square$

**Lemma B.2** *Let Assumptions 1.2, 1.3, 2.1 hold and let $x \in \mathcal{X}$. For every integer $T \geq 1$, we have*

$$\sum_{k=1}^{T} \lambda_k \langle F(x_k), x_k - x \rangle \leq \tfrac{1}{2} \|x - x_0\|^2, \qquad \sum_{k=1}^{T} \|x_k - v_k\|^2 \leq 4\|x^* - x_0\|^2, \qquad (38)$$

*where $x^* \in \mathcal{X}$ denotes the weak solution to the VI.*

*Proof.* For any $x \in \mathcal{X}$, we have

$$\mathcal{E}_0 - \mathcal{E}_T + \langle s_T, x - x_0 \rangle = \mathcal{E}_0 - \left( \max_{v \in \mathcal{X}} \langle s_T, v - x_0 \rangle - \tfrac{1}{2} \|v - x_0\|^2 \right) + \langle s_T, x - x_0 \rangle.$$

Since $s_0 = 0_d$, we have $\mathcal{E}_0 = 0$ and

$$\mathcal{E}_0 - \mathcal{E}_T + \langle s_T, x - x_0 \rangle \leq - \left( \langle s_T, x - x_0 \rangle - \tfrac{1}{2}\|x - x_0\|^2 \right) + \langle s_T, x - x_0 \rangle = \tfrac{1}{2}\|x - x_0\|^2.$$

This together with Lemma B.1 yields that

$$\sum_{k=1}^{T} \lambda_k \langle F(x_k), x_k - x \rangle + \tfrac{1}{8} \left( \sum_{k=1}^{T} \|x_k - v_k\|^2 \right) \leq \tfrac{1}{2}\|x - x_0\|^2, \quad \text{for all } x \in \mathcal{X},$$

which implies the first inequality. Since the VI satisfies the Minty condition, there exists $x^* \in \mathcal{X}$ such that $\langle F(x_k), x_k - x^* \rangle \geq 0$ for all $k \geq 1$. Letting $x = x^*$ in the above inequality yields the second inequality. $\qquad\square$

**Lemma B.3** *Let Assumptions 1.2, 1.3, 2.1 hold. For every integer $T \geq 1$, we have*

$$\frac{1}{\left( \sum_{k=1}^{T} \lambda_k \right)^2} \leq \frac{2048 L^2 \|x^* - x_0\|^2}{T^3} + \frac{2048 \delta^2}{T^2} \qquad (39)$$

*where $x^* \in \mathcal{X}$ denotes the weak solution to the VI.*

*Proof.* Without loss of generality, we assume that $x_0 \neq x^*$. We have

$$\sum_{k=1}^{T} (\lambda_k)^{-2} (\tfrac{1}{32})^2 \leq \sum_{k=1}^{T} (\lambda_k)^{-2} \left( \lambda_k \left( \tfrac{L}{2} \|x_k - v_k\| + \delta \right) \right)^2 = \sum_{k=1}^{T} \left( \tfrac{L}{2} \|x_k - v_k\| + \delta \right)^2$$

$$\leq \sum_{k=1}^{T} \tfrac{L^2}{2} \|x_k - v_k\|^2 + 2T\delta^2 \overset{\text{Lemma B.2}}{\leq} 2L^2 \|x^* - x_0\|^2 + 2T\delta^2.$$

By the Hölder inequality, we have

$$\sum_{k=1}^{T} 1 = \sum_{k=1}^{T} \left( (\lambda_k)^{-2} \right)^{1/3} (\lambda_k)^{2/3} \leq \left( \sum_{k=1}^{T} (\lambda_k)^{-2} \right)^{1/3} \left( \sum_{k=1}^{T} \lambda_k \right)^{2/3}.$$

Putting these pieces together yields that

$$T \leq 32^{2/3} (2L^2 \|x^* - x_0\|^2 + 2\delta^2 T)^{\frac{1}{3}} \left( \sum_{k=1}^{T} \lambda_k \right)^{2/3},$$

Plugging this into the above inequality yields that

$$\frac{1}{\left( \sum_{k=1}^{T} \lambda_k \right)^2} \leq \frac{2048 L^2 \|x^* - x_0\|^2}{T^3} + \frac{2048 \delta^2}{T^2}$$

$\qquad\square$

**Theorem 3.2** *Let Assumptions 1.2, 1.1, 2.1. Then, after $T \geq 1$ iterations of VIJI with parameters $\beta = \delta$, $\eta = 10$, $\mathbf{opt} = 0$, we get the following bound*

$$\text{GAP}(\tilde{x}_T) = \sup_{x \in \mathcal{X}} \langle F(x), \tilde{x}_T - x \rangle \leq \tfrac{16\sqrt{2}LD^3}{T^{3/2}} + \tfrac{16\sqrt{2}\delta D^2}{T}.$$

*Proof.* Letting $x \in \mathcal{X}$, we derive from the monotonicity of $F$ and the definition of $\tilde{x}_T$ (i.e., $\mathsf{opt} = 0$) that

$$\langle F(x), \tilde{x}_T - x \rangle = \tfrac{1}{\sum_{k=1}^{T} \lambda_k} \left( \sum_{k=1}^{T} \lambda_k \langle F(x), x_k - x \rangle \right).$$

Combining this inequality with the first inequality in Lemma B.2 yields that

$$\langle F(x), \tilde{x}_T - x \rangle \leq \tfrac{\|x - x_0\|^2}{2(\sum_{k=1}^{T} \lambda_k)}, \quad \text{for all } x \in \mathcal{X}.$$

Since $x_0 \in \mathcal{X}$, we have $\|x - x_0\| \leq D$ and hence

$$\langle F(x), \tilde{x}_T - x \rangle \leq \tfrac{D^2}{2(\sum_{k=1}^{T} \lambda_k)}, \quad \text{for all } x \in \mathcal{X}.$$

Then, we combine Lemma B.3 and the fact that $\|x^* - x_0\| \leq D$ to obtain that

$$\langle F(x), \tilde{x}_T - x \rangle \leq \frac{D^2}{2} \sqrt{\frac{2048 L^2 D^2}{T^3} + \frac{2048 \delta^2}{T^2}} \leq \frac{16\sqrt{2} L D^3}{T^{3/2}} + \frac{16\sqrt{2} \delta D^2}{T}, \quad \text{for all } x \in \mathcal{X}.$$

By the definition of a gap function. (4), we have

$$\mathrm{GAP}(\tilde{x}_T) = \sup_{x \in \mathcal{X}} \langle F(x), \tilde{x}_T - x \rangle \leq \tfrac{16\sqrt{2} L D^3}{T^{3/2}} + \tfrac{16\sqrt{2} \delta D^2}{T}. \tag{40}$$

$\square$

## B.2 Proof of Theorem 3.3

We directly follow the steps of the proof of Theorem 3.2. Lemmas B.1, B.2 remain the same. Because of the choice of $\beta_{k+1} = \frac{L_1}{2}\|x_{k+1} - v_{k+1}\|$ adaptive strategy for $\lambda_{k+1}$ looks as follows: $\frac{1}{32} \leq L\|x_{k+1} - v_{k+1}\| \leq \frac{1}{22}$. Next Lemma is a counterpart of Lemma B.3.

**Lemma B.4** *Let Assumptions 1.2, 1.3, 2.1 hold. For every integer $T \geq 1$, we have*

$$\frac{1}{\sum_{k=1}^{T} \lambda_k} \leq \frac{64 L \|x^* - x_0\|}{T^{3/2}} \tag{41}$$

*where $x^* \in \mathcal{X}$ denotes the weak solution to the VI.*

*Proof.* Without loss of generality, we assume that $x_0 \neq x^*$. We have

$$\sum_{k=1}^{T} (\lambda_k)^{-2} (\tfrac{1}{32})^2 \leq \sum_{k=1}^{T} (\lambda_k)^{-2} \left( \lambda_k \left( L\|x_k - v_k\| \right) \right)^2 \leq \sum_{k=1}^{T} L^2 \|x_k - v_k\|^2 \overset{\text{Lemma } B.2}{\leq} 4 L^2 \|x^* - x_0\|^2.$$

By the Hölder inequality, we have

$$\sum_{k=1}^{T} 1 = \sum_{k=1}^{T} \left( (\lambda_k)^{-2} \right)^{1/3} (\lambda_k)^{2/3} \leq \left( \sum_{k=1}^{T} (\lambda_k)^{-2} \right)^{1/3} \left( \sum_{k=1}^{T} \lambda_k \right)^{2/3}.$$

Putting these pieces together yields that

$$T \leq 32^{2/3} (4 L^2 \|x^* - x_0\|^2)^{\frac{1}{3}} \left( \sum_{k=1}^{T} \lambda_k \right)^{2/3}.$$

Plugging this into the above inequality yields that

$$\frac{1}{\sum_{k=1}^{T} \lambda_k} \leq \frac{64 L \|x^* - x_0\|}{T^{3/2}}.$$

$\square$

Then, by following the rest of the proof of Theorem 3.2, we get

**Theorem 3.3** *Let Assumptions 1.2, 1.1 hold. Let $\{x_k, v_k\}$ be iterates generated by Algorithm 1 and*

$$\|(\nabla F(v_k) - J(v_k))[z_k - v_k]\| \leq \delta_k \|z_k - v_k\|, \quad \delta_k \leq \tfrac{L_1}{2}\|x_k - v_k\|.$$

*Then, after $T \geq 1$ iterations of Algorithm 1 with parameters $\beta_k = \frac{L_1}{2}\|x_k - v_k\|$, $\eta = 10$, $\mathsf{opt} = 0$, we get the following bound*

$$\mathrm{GAP}(\tilde{x}_T) = \sup_{x \in \mathcal{X}} \langle F(x), \tilde{x}_T - x \rangle \leq \tfrac{32 L D^3}{T^{3/2}}.$$

# C Proofs of Theorems 3.4, 3.5

In this section, we let $L := L_1$.

**Theorem 3.4** *Let Assumptions 1.2, 1.3, 2.1 hold. Then after $T \geq 1$ iterations of Algorithm 1 with parameters $\beta_k = \delta, \eta = 10, \mathsf{opt} = 2$ we get the following bound*

$$\mathrm{RES}(\hat{x}) := \sup_{x \in \mathcal{X}} \langle F(\hat{x}_T), \hat{x}_T - x \rangle = O\left(\frac{LD^3}{T} + \frac{\delta D^2}{\sqrt{T}}\right). \tag{42}$$

*Proof.* Since we consider $\mathsf{opt} = 2$, then

$$\mathrm{RES}(\hat{x}) = \mathrm{RES}(x_{k_T}) = \sup_{x \in \mathcal{X}} \langle F(x_{k_T}), x_{k_T} - x \rangle.$$

From (36) we get

$$\langle F(x_k), x_k - x \rangle = \langle F(x_k) - \Omega_{v_k}^\eta(x_k), x_k - x \rangle + \langle \Omega_{v_k}^\eta(x_k), x_k - x \rangle$$
$$\overset{(36)}{\leq} \|F(x_k) - \Omega_{v_k}^\eta(x_k)\| \|x_k - x\| + \frac{L}{2}\|x_k - v_k\|^3 + \delta\|x_k - v_k\|^2. \tag{43}$$

Next, from triangle inequality we have

$$\|F(x_k) - \Psi_{v_k}(x_k)\| = \left\| F(x_k) - \Omega_{v_k}^\eta(x_k) + \eta\delta(x_k - v_k) + \frac{5L}{2}\|x_k - v_k\|(x_k - v_k) \right\|$$

$$\geq \|F(x_k) - \Omega_{v_k}^\eta(x_k)\| - \eta\delta\|x_k - v_k\| - \frac{5L}{2}\|x_k - v_k\|^2.$$

From this and (29) we get

$$\|F(x_k) - \Omega_{v_k}^\eta(x_k)\| \leq \frac{L}{2}\|x_k - v_k\|^2 + \delta\|x_k - v_k\| + \eta\delta\|x_k - v_k\| + \frac{5L}{2}\|x_k - v_k\|^2$$
$$= 3L\|x_k - v_k\|^2 + \delta(\eta + 1)\|x_k - v_k\|.$$

Now, we can return to (43):

$$\langle F(x_k), x_k - x \rangle$$
$$\overset{(43),(44)}{\leq} 3L\|x_k - v_k\|^2\|x_k - x\| + \delta(\eta + 1)\|x_k - v_k\|\|x_k - x\| + \frac{L}{2}\|x_k - v_k\|^3 + \delta\|x_k - v_k\|^2$$
$$= L\|x_k - v_k\|^2\left(3\|x_k - x\| + \frac{1}{2}\|x_k - v_k\|\right) + \delta\|x_k - v_k\|\left((\eta + 1)\|x_k - x\| + \|x_k - v_k\|\right).$$

Since $D := \max_{x,y \in \mathcal{X}} \|x - y\|$,

$$\langle F(x_k), x_k - x \rangle \leq \frac{7}{2}LD\|x_k - v_k\|^2 + \delta(\eta + 2)\|x_k - v_k\|. \tag{44}$$

Next, from second inequality from (38) and from definition of $x_{k_T}$ in Algorithm 1 we obtain

$$\|x_{k_T} - v_{k_T}\|^2 \equiv \min_{1 \leq k \leq T}\|x_k - v_k\|^2 \overset{(38)}{\leq} \frac{1}{T}\sum_{k=1}^T \|x_k - v_k\|^2 \leq \frac{4\|x^* - x_0\|^2}{T}. \tag{45}$$

Since (44) holds for any $x \in \mathcal{X}$, we get final result

$$\mathrm{RES}(x_{k_T}) := \sup_{x \in \mathcal{X}} \langle F(x_{k_t}), x_{k_T} - x \rangle \overset{(44),(45)}{\leq} \frac{14LD\|x^* - x_0\|^2}{T} + \frac{2\delta(\eta + 2)D\|x^* - x_0\|}{\sqrt{T}}$$
$$\leq \frac{14LD^3}{T} + \frac{2\delta(\eta + 2)D^2}{\sqrt{T}} = \frac{14LD^3}{T} + \frac{24D^2}{\sqrt{T}}.$$

$\square$

**Theorem 3.5** *Let Assumptions 1.2, 1.3 hold. Let $\{x_k, v_k\}$ be iterates generated by Algorithm 1 that satisfy (14). Then after $T \geq 1$ iterations of Algorithm 1 with parameters $\beta_k = \frac{L}{2}\|x_k - v_k\|$, $\eta = 10, \mathsf{opt} = 2$ we get the following bound*

$$\mathrm{RES}(\hat{x}) := \sup_{x \in \mathcal{X}} \langle F(\hat{x}_T), \hat{x}_T - x \rangle = \frac{38L_1 D^3}{T}. \tag{46}$$

The proof of this theorem repeats the proof of Theorem 3.4 with $\delta \leq \frac{L}{2}\|x_k - v_k\|$.

# D   Proof of Theorem 4.3

**Theorem 4.3** *Let some first-order method $\mathcal{M}$ satisfy Assumption 4.1 and have access only $\delta$-inexact first-order oracle 15. Assume the method $\mathcal{M}$ ensures for any $L_0$-zero-order smooth and $L_1$-first-order smooth monotone operator $F$ the following convergence rate*

$$\mathrm{GAP}(\hat{x}) \leq O(1) \max\left\{\tfrac{\delta D^2}{\Xi_1(T)}; \tfrac{L_1 D^3}{\Xi_2(T)}\right\}.$$

*Then for all $T \geq 1$ we have*

$$\Xi_1(T) \leq T, \qquad \Xi_2(T) \leq T^{3/2}.$$

*Proof.* We prove this Theorem by contradiction. Assume the existence of a method $\mathcal{M}$ that satisfies the conditions of Theorem 4.3 and achieves faster rate in one of the terms (18).
First, suppose $\Xi_1(T) > T$. Consider the first-order lower bound from [89], which is established using a quadratic min-max problem as the worst-case function. In this scenario, the operator has a $0$-Lipschitz continuous Jacobian. Applying first-order method $\mathcal{M}$ to this lower bound and using an inexact Jacobian $J(x) = L_0 I_{d \times d}$, yields the rate $O\left(\frac{L_0 D^2}{\Xi_1(T)}\right)$, $\Xi_1(T) > T$, which is faster than the lower bound $\Omega\left(\frac{L_0 D^2}{T}\right)$, contradicting our assumption.
Secondly, let us assume $\Xi_2(T) > T^{3/2}$. The lower bound for exact second-order methods is $\Omega\left(\frac{L_1 D^3}{T^{3/2}}\right)$ [69]. By taking exact Jacobian in the method $\mathcal{M}$ ($\delta = 0$), we place $\mathcal{M}$ in the class of exact second-order methods. Consequently, we obtain a contradiction with the lower bound. $\qquad\square$

# E   Proof of Theorem 5.1

*Proof.* The proof is common for both formulas (20) and (21), where $\alpha = 1$ for classical L-Broyden approximation and $\alpha = m + 1$ for Damped L-Broyden approximation. First, from $L_0$-zero-order smoothness, get

$$\|\nabla F(x) - J_x\|_{\mathrm{op}} \leq \|\nabla F(x)\|_{op} + \|J_x\|_{op} \leq L_0 + \|J_x\|_{op}$$

Now, we upper-bound $\|J_x\|_{op} = \|J^m\|_{op}$ by induction:

$$\left\|J^{i+1}\right\|_{op} \leq \left\|J^i + \tfrac{(y_i - J^i s_i)s_i^\top}{\alpha s_i^\top s_i}\right\|_{op} \leq \left\|J^i\left(I - \tfrac{s_i s_i^\top}{\alpha s_i^\top s_i}\right) + \tfrac{y_i s_i^\top}{\alpha s_i^\top s_i}\right\|_{op}$$

$$\leq \left\|J^i\left(I - \tfrac{s_i s_i^\top}{\alpha s_i^\top s_i}\right)\right\|_{op} + \left\|\tfrac{y_i s_i^\top}{\alpha s_i^\top s_i}\right\|_{op} \leq \|J^i\|_{op}\left\|I - \tfrac{s_i s_i^\top}{\alpha s_i^\top s_i}\right\|_{op} + \tfrac{\|y_i\|\|s_i\|}{\alpha s_i^\top s_i} \leq \|J^i\|_{op} + \tfrac{L_0}{\alpha},$$

where the last inequality is coming from $L_0$-zero-order smoothness for operator difference or JVP. By summing up the previous inequality for $i$ in $0, \ldots, m - 1$, we get $\|J^m\|_{op} \leq \|J^0\|_{op} + \frac{m L_0}{\alpha}$. Finally, we prove the result of Theorem 5.1. $\qquad\square$

# F   Proof of Theorem 6.2

In this section, we let $L := L_1$.

**Theorem 6.2** *Let Assumptions 1.2, 2.1, 6.1 hold. Then the total number of iterations of Algorithm 2 to reach desired accuracy $\|z_s - x^*\| \leq \varepsilon$ is*

$$O\left(\left(\tfrac{LD}{\mu}\right)^{\frac{2}{3}} + \left(\tfrac{\delta}{\mu} + 1\right)\lceil\log\tfrac{D}{\varepsilon}\rceil\right).$$

*Proof.* From the definition of $\tilde{x}_T$, Jensen inequality, strong monotonicity (25), definition of strong minty problem (2) and Lemmas B.2, B.3 we get

$$\mu\|\tilde{x}_T - x^*\|^2 = \mu \left\| \frac{1}{\sum_{k=1}^T \lambda_t} \sum_{k=1}^T (\lambda_k x_k - \lambda_k x^*) \right\|^2$$

$$\leq \frac{\mu}{\sum_{k=1}^T \lambda_t} \sum_{k=1}^T \lambda_k \|x_k - x^*\|^2$$

$$\overset{(25)}{\leq} \frac{1}{\sum_{k=1}^T \lambda_t} \sum_{k=1}^T \lambda_k \langle F(x_k) - F(x^*), x_k - x^* \rangle$$

$$\overset{(2)}{\leq} \frac{1}{\sum_{k=1}^T \lambda_t} \sum_{k=1}^T \lambda_k \langle F(x_k), x_k - x^* \rangle$$

$$\overset{(38),(39)}{\leq} \left( \frac{2048 L^2 \|x_0 - x^*\|^2}{T^3} + \frac{2048\delta^2}{T^2} \right)^{\frac{1}{2}} \cdot \frac{1}{2} \|x_0 - x^*\|^2$$

$$\leq \left( 2\max\left\{ \frac{2048 L^2 \|x_0 - x^*\|^2}{T_i^3}, \frac{2048\delta^2}{T_i^2} \right\} \right)^{\frac{1}{2}} \cdot \frac{1}{2} \|x_0 - x^*\|^2.$$

Denote $R := D$, $R_i = \frac{R}{2^i}$, $i \geq 1$. Now we run Algorithm 1 in cycle for $i = 1, ..., n$ and restart it every time its distance to the solution becomes at least twice less than $R_{i-1}$. Thus, let $T_i$ be number of iterations we run Algorithm 1 inside cycle of Algorithm 2. In other words, let $T_i$ be such that $\left\|\tilde{x}_{T_{i-1}} - x^*\right\| \leq \frac{R_{i-1}}{2}$ where $\tilde{x}_{T_{i+1}}$ is the point, where we restart Algorithm 1. Then the number of iterations before the $i$-th restart is

$$\mu\|\tilde{x}_{T_i} - x^*\|^2 \leq \left( 2\max\left\{ \frac{2048 L^2 R_{i-1}^2}{T_i^3}, \frac{2048\delta^2}{T_i^2} \right\} \right)^{\frac{1}{2}} \cdot \frac{1}{2} R_{i-1}^2$$

$$\leq \frac{\mu R_{i-1}^2}{4} \Leftrightarrow$$

$$\Leftrightarrow \left( \max\left\{ \frac{2048 L^2 R_{i-1}^2}{T_i^3}, \frac{2048\delta^2}{T_i^2} \right\} \right)^{\frac{1}{2}} \leq \frac{\mu}{2\sqrt{2}}.$$

(47)

Deriving $T_i$ for each case under $\max$, we get

$$\begin{cases} T_i \geq \dfrac{2^{\frac{14}{3}} L^{\frac{2}{3}} R_{i-1}^{\frac{2}{3}}}{\mu^{\frac{2}{3}}} \\ T_i \geq \dfrac{2^7 \delta}{\mu}. \end{cases}$$

Thus,

$$T_i = \left\lceil \max\left\{ \frac{2^{\frac{14}{3}} L^{\frac{2}{3}} R_{i-1}^{\frac{2}{3}}}{\mu^{\frac{2}{3}}}, \frac{2^7 \delta}{\mu} \right\} \right\rceil$$

(48)

Now we calculate the total number of restarts to reach $\|\tilde{x}_{T_n} - x^*\| \leq \varepsilon$. From (47) we can get that

$$\|\tilde{x}_{T_n} - x^*\| \leq \sqrt{ \frac{1}{\mu} \left( 2\max\left\{ \frac{2048 L^2 R_{n-1}^2}{T_n^3}, \frac{2048\delta^2}{T_n^2} \right\} \right)^{\frac{1}{2}} \cdot \frac{1}{2} R_{n-1}^2 }$$

$$\overset{(48)}{\leq} \frac{R_{n-1}}{\sqrt{2\mu}} \left( 2\max\left\{ \frac{\mu^2}{8}, \frac{\mu^2}{8} \right\} \right)^{\frac{1}{4}}$$

$$= \frac{R_{n-1}}{2}$$

$$= R \cdot 2^{-(n-1)-1} \leq \varepsilon.$$

Deriving $n$ from last inequality, we get

$$n = \left\lceil \log \frac{R}{\varepsilon} \right\rceil. \tag{49}$$

Now we provide auxiliary estimation, that we will need next:

$$
\begin{aligned}
\sum_{i=1}^{n} R_i^{\frac{2}{3}} &= \sum_{i=1}^{n} \frac{\|x_0 - x^*\|^{\frac{2}{3}}}{2^{\frac{2(i-1)}{3}}} \\
&= \|x_0 - x^*\|^{\frac{2}{3}} \frac{1 - 2^{\frac{-2(n-1)}{3}}}{2^{\frac{1}{3}}} \\
&\leq \|x_0 - x^*\|^{\frac{2}{3}} 2^{-\frac{1}{3}} \\
&\leq 2^{-\frac{1}{3}} D^{\frac{2}{3}}.
\end{aligned}
\tag{50}
$$

Finally, we can get the total number of iterations of Algorithm 1 inside Algorithm 2:

$$
\begin{aligned}
\sum_{i=1}^{n} T_i &\overset{(48)}{=} \sum_{i=1}^{n} \left\lceil \max\left\{ \frac{2^{\frac{14}{3}} L^{\frac{2}{3}} R_{i-1}^{\frac{2}{3}}}{\mu^{\frac{2}{3}}}, \frac{2^7 \delta}{\mu} \right\} \right\rceil \\
&\leq \sum_{i=1}^{n} \frac{2^{\frac{14}{3}} L^{\frac{2}{3}} R_{i-1}^{\frac{2}{3}}}{\mu^{\frac{2}{3}}} + \frac{2^7 \delta}{\mu} n + n \\
&= \frac{2^{\frac{14}{3}} L^{\frac{2}{3}}}{\mu^{\frac{2}{3}}} \sum_{i=1}^{n} R_{i-1}^{\frac{2}{3}} + \frac{2^7 \delta}{\mu} n + n \\
&\overset{(50),(49)}{\leq} \frac{2^{\frac{13}{3}} L^{\frac{2}{3}} D^{\frac{2}{3}}}{\mu^{\frac{2}{3}}} + \left( \frac{2^7 \delta}{\mu} + 1 \right) \left\lceil \log \frac{D}{\varepsilon} \right\rceil
\end{aligned}
$$

Thus,

$$\sum_{i=1}^{n} T_i = O\left( \left( \frac{LD}{\mu} \right)^{\frac{2}{3}} + \left( \frac{\delta}{\mu} + 1 \right) \left\lceil \log \frac{D}{\varepsilon} \right\rceil \right).$$

This completes the proof. $\qquad \square$

# G   Tensor generalization with more details

## G.1   Preliminaries

---
**Algorithm 3** VIHI
---

**Input:** initial point $x_0 \in \mathcal{X}$, parameters $L_1, \eta$, sequence $\{\delta_i\}_{i=1}^{p-1}$, and opt $\in \{0, 1, 2\}$.

**Initialization:** set $s_0 = 0 \in \mathbb{R}^d$.

**for** $k = 0, 1, 2, \ldots, T$ **do**

    Compute $v_{k+1} = \mathrm{argmax}_{v \in \mathcal{X}} \{\langle s_k, v - x_0 \rangle - \frac{1}{2}\|v - x_0\|^2\}$.

    Compute $x_{k+1} \in \mathcal{X}$ such that condition (53) holds true.

    Compute $\lambda_{k+1}$ such that

$$\tfrac{1}{4(5p-2)} \leq \lambda_k \left( \tfrac{L_{p-1}}{p!} \|x_k - v_k\|^{p-1} + \sum_{i=1}^{p-1} \tfrac{\delta_i}{i!} \|x_k - v_k\|^{i-1} \right) \leq \tfrac{1}{2(5p+1)}.$$

    Compute $s_{k+1} = s_k - \lambda_{k+1} F(x_{k+1})$.

**Output:** $\hat{x} = \begin{cases} \tilde{x}_T = \frac{1}{\sum_{k=1}^{T} \lambda_k} \sum_{k=1}^{T} \lambda_k x_k, & \text{if opt} = 0, \\ x_T, & \text{else if opt} = 1, \\ x_{k_T} \text{ for } k_T = \mathrm{argmin}_{1 \leq k \leq T} \|x_k - v_k\|, & \text{else if opt} = 2. \end{cases}$

---

In this section, we provide more details on the generalization of Algorithm 1 with high-order derivatives. We provide the pseudocode of the resulting method in Algorithm 3. To show the convergence of Algorithm 3, we use the Lyapunov function (30).

Define the $(p-1)$-th order approximations of the $F$

$$\Phi_{p,v}(x) = F(v) + \sum_{i=1}^{p-1} \frac{1}{i!} \nabla^i F(v)[x-v]^i \tag{51}$$

$$\Psi_{p,v}(x) = F(v) + \sum_{i=1}^{p-1} \frac{1}{i!} G_i(v)[x-v]^i. \tag{52}$$

On each step, our method solves the following subproblem:

$$\sup_{x \in \mathcal{X}} \langle \Omega_{p,v_k}(x_k), x_k - x \rangle \leq \frac{L_{p-1}}{p!} \|x_k - v_k\|^{p+1} + \sum_{i=1}^{p-1} \frac{\delta_i}{i!} \|x_k - v_k\|^{i+1}. \tag{53}$$

Additionally, we will need an auxiliary result from [52], based on Assumption 4.2. The authors show, that this Assumption allows to control the quality of approximation of operator $F$ by its high-order Taylor polynomial:

$$\|F(v) - \Phi_{p,v}(x)\| \leq \frac{L_{p-1}}{p!} \|x-v\|^p. \tag{54}$$

## G.2   Auxiliary lemmas

First of all, we provide high-order generalizations of auxiliary lemmas for second-order case from Section A.

**Lemma G.1** *Let Assumptions 7.1 and 4.2 with $i = p-1$ hold. Then, for any $x, v \in \mathcal{X}$*

$$\|F(v) - \Psi_{p,v}(x)\| \leq \frac{L_{p-1}}{p!} \|x-v\|^p + \sum_{i=1}^{p-1} \frac{1}{i!} \delta_i \|x-v\|^i. \tag{55}$$

*Proof.*

$$\|F(v) - \Psi_{p,v}(x)\| \leq \|F(v) - \Phi_{p,v}(x)\| + \|\Phi_{p,v}(x) - \Psi_{p,v}(x)\|$$

$$\overset{(54)}{\leq} \frac{L_{p-1}}{p!} \|x-v\|^p + \sum_{i-1}^{p-1} \frac{1}{i!} \|(\nabla^i F(v) - G_i(v))[x-v]^{i-1}\| \|x-v\|$$

$$\overset{(27)}{\leq} \frac{L_{p-1}}{p!} \|x-v\|^p + \sum_{i=1}^{p-1} \frac{\delta_i}{i!} \|x-v\|^i.$$

$\square$

**Lemma G.2** *Let Assumptions 1.1, 7.1 and 4.2 with $i = p-1$ hold. Then for any $x, v_{k+1} \in \mathcal{X}$ VI (11) is relatively strongly monotone if $\eta_i \geq p$*

$$\frac{1}{2} \left( \nabla \Omega_{p,v}(x) + \nabla \Omega_{p,v}(x)^T \right)$$

$$\frac{4L_{p-1}}{(p-1)!} \left( \|x-v\|^{p-1} I_{n \times n} + \|x-v\|^{p-3}(x-v)(x-v)^T \right)$$

$$+ \sum_{i=1}^{p-1} \frac{\delta_i}{i!} \|x-v\|^{i-3} \left( (\eta_i - i)\|x-v\|^2 I_{n \times n} + \eta_i(i-1)(x-v)(x-v)^T \right).$$

*Proof.*

$$\frac{1}{2}\left(\nabla\Omega_{p,v}(x) + \nabla\Omega_{p,v}(x)^T\right)$$

$$\stackrel{(28)}{=} \frac{1}{2}\left(\sum_{i=1}^{p-1}\frac{1}{(i-1)!}\left(G_i(v)[x-v]^{i-1} + \left(G_i(v)[x-v]^{i-1}\right)^T\right)\right)$$

$$+\sum_{i=1}^{p-1}\frac{\eta_i\delta_i}{i!}\left(\|x-v\|^{i-1}I_{n\times n} + (i-1)\|x-v\|^{i-3}(x-v)(x-v)^T\right)$$

$$+\frac{5L_{p-1}}{(p-1)!}\left(\|x-v\|^{p-1}I_{n\times n}\right)$$

$$\stackrel{(55)}{\succeq} \frac{1}{2}\left(\nabla F(x) + \nabla F(x)^T\right)$$

$$-\frac{L_{p-1}}{(p-1)!}\|x-v\|^{p-1}I_{n\times n} - \sum_{i=1}^{p-1}\frac{1}{(i-1)!}\delta_i\|x-v\|^{i-1}I_{n\times n}$$

$$+\sum_{i-1}^{p-1}\frac{\eta_i\delta_i}{i!}\left(\|x-v\|^{i-1}I_{n\times n} + (i-1)\|x-v\|^{i-3}(x-v)(x-v)^T\right)$$

$$+\frac{5L_{p-1}}{(p-1)!}\left(\|x-v\|^{p-1}I_{n\times n} + (p-1)\|x-v\|^{p-3}(x-v)(x-v)^T\right)$$

$$= \frac{1}{2}\left(\nabla F(x) + \nabla F(x)^T\right)$$

$$+\frac{4L_{p-1}}{(p-1)!}\|x-v\|^{p-1}I_{n\times n} + \frac{5L_{p-1}}{(p-2)!}\|x-v\|^{p-3}(x-v)(x-v)^T$$

$$+\sum_{i=1}^{p-1}\frac{\delta_i}{i!}(\eta_i - i)\|x-v\|^{i-1}I_{n\times n} + \sum_{i=1}^{p-1}\frac{\eta_i\delta_i(i-1)}{i!}\|x-v\|^{i-3}(x-v)(x-v)^T.$$

From monotonicity of $F$ we know that $\frac{1}{2}(F(x) - F(x)^T) \succeq 0$. Thus,

$$\frac{1}{2}\left(\nabla\Omega_{p,v}(x) + \nabla\Omega_{p,v}(x)^T\right)$$

$$\succeq \frac{4L_{p-1}}{(p-1)!}\|x-v\|^{p-1}I_{n\times n} + \frac{5L_{p-1}}{(p-2)!}\|x-v\|^{p-3}(x-v)(x-v)^T$$

$$+\sum_{i=1}^{p-1}\frac{\delta_i}{i!}(\eta_i - i)\|x-v\|^{i-1}I_{n\times n} + \sum_{i=1}^{p-1}\frac{\eta_i\delta_i(i-1)}{i!}\|x-v\|^{i-3}(x-v)(x-v)^T.$$

By rearranging the terms we get

$$\frac{1}{2}\left(\nabla\Omega_{p,v}(x) + \nabla\Omega_{p,v}(x)^T\right)$$

$$\frac{4L_{p-1}}{(p-1)!}\left(\|x-v\|^{p-1}I_{n\times n} + \|x-v\|^{p-3}(x-v)(x-v)^T\right)$$

$$+\sum_{i=1}^{p-1}\frac{\delta_i}{i!}\|x-v\|^{i-3}\left((\eta_i - i)\|x-v\|^2 I_{n\times n} + \eta_i(i-1)(x-v)(x-v)^T\right).$$

Thus, if $\eta_i \geq p$, we get that $\frac{1}{2}\left(\nabla\Omega_{p,v_{k+1}}(x) + \nabla\Omega_{p,v_{k+1}}(x)^T\right)$ is relatively strongly monotone. $\square$

### G.3 Convergence in monotone case

In this subsection we provide theoretical results directly connected to convergence rate of Algorithm 3. Firstly, we need to introduce additional technical lemmas, that generalize corresponding lemmas in Section B.

**Lemma G.3** *Let Assumption 1.1 hold and $\eta_i = 5p$. Then, for every $T \geq 1$, we have*

$$\sum_{k=1}^{T}\langle\lambda_k F(x_k), x_0 - x\rangle \leq \mathcal{E}_0 - \mathcal{E}_T + \langle s_t, x - x_0\rangle - \frac{1}{8}\sum_{k=1}^{T}\|x_k - v_k\|^2. \tag{56}$$

*Proof.* Since the first steps in the proof of this Lemma are the same as in Lemma B.1, we start our reasoning from (33):

$$\sum_{k=1}^{T} \lambda_k \langle F(x_k), x_k - x \rangle \leq \mathcal{E}_0 - \mathcal{E}_T + \underbrace{\sum_{k=1}^{T} \lambda_k \langle F(x_k), x_0 - x \rangle}_{\mathbf{I}} + \underbrace{\sum_{k=1}^{T} \lambda_k \langle F(x_k), x_k - v_{k+1} \rangle - \frac{1}{2} \|v_k - v_{k+1}\|^2}_{\mathbf{II}}.$$

$$(57)$$

Using the update formula for $s_{k+1}$ and letting $s_0 = 0_d \in \mathbb{R}^d$, we have

$$\mathbf{I} = \sum_{k=1}^{T} \langle \lambda_k F(x_k), x_0 - x \rangle = \sum_{k=1}^{T} \langle s_{k-1} - s_k, x_0 - x \rangle = \langle s_T, x - x_0 \rangle. \qquad (58)$$

Now consider **II**. Using (28) we get

$$\langle F(x_k), x_k - v_{k+1} \rangle$$

$$\overset{(28)}{=} \langle F(x_k) - \Psi_{p,v_k}(x_k), x_k - v_{k+1} \rangle + \langle \Omega_{p,v_k}(x_k), x_k - v_{k+1} \rangle$$

$$- \sum_{i=1}^{p-1} \frac{\eta_i \delta_i}{i!} \|x_k - v_k\|^{i-1} \langle x_k - v_k, x_k - v_{k+1} \rangle$$

$$- \frac{5 L_{p-1}}{(p-1)!} \|x_k - v_k\|^{p-1} \langle x_k - v_k, x_k - v_{k+1} \rangle$$

$$\overset{(55)}{\leq} \frac{L_{p-1}}{p!} \|x_k - v_k\|^{p} \|x_k - v_{k+1}\| + \sum_{i=1}^{p-1} \frac{1}{i!} \delta_i \|x_k - v_k\|^{i} \|x_k - v_{k+1}\| + \langle \Omega_{p,v_k}(x_k), x_k - v_{k+1} \rangle$$

$$- \sum_{i=1}^{p-1} \frac{\eta_i \delta_i}{i!} \|x_k - v_k\|^{i-1} \langle x_k - v_k, x_k - v_{k+1} \rangle$$

$$- \frac{5 L_{p-1}}{(p-1)!} \|x_k - v_k\|^{p-1} \langle x_k - v_k, x_k - v_{k+1} \rangle.$$

Next, using $\langle x_k - v_k, x_k - v_{k+1} \rangle \geq \|x_k - v_k\|^2 - \|x_k - v_k\|\|v_k - v_{k+1}\|$ and $\|x_k - v_{k+1}\| \leq \|x_k - v_k\| + \|v_k - v_{k+1}\|$, we get

$$\langle F(x_k), x_k - v_{k+1} \rangle$$

$$\leq \frac{L_{p-1}}{p!} \left( \|x_k - v_k\|^{p+1} + \|x_k - v_k\|^{p} \|v_k - v_{k+1}\| \right)$$

$$+ \sum_{i=1}^{p-1} \frac{1}{i!} \delta_i \left( \|x_k - v_k\|^{i+1} \|x_k - v_k\|^{i} \|v_k - v_{k+1}\| \right)$$

$$- \sum_{i=1}^{p-1} \frac{\eta_i \delta_i}{i!} \left( \|x_k - v_k\|^{i+1} - \|x_k - v_k\|^{i} \|v_k - v_{k+1}\| \right)$$

$$- \frac{5 L_{p-1}}{(p-1)!} \left( \|x_k - v_k\|^{p+1} - \|x_k - v_k\|^{p} \|v_k - v_{k+1}\| \right)$$

$$+ \langle \Omega_{p,v_k}(x_k), x_k - v_{k+1} \rangle.$$

From definition of the subproblem (53) we have that

$$\langle \Omega_{p,v_k}(x_k), x_k - v_{k+1} \rangle \leq \sup_{x \in \mathcal{X}} \langle \Omega_{p,v_k}(x_k), x_k - x \rangle \leq \frac{L_{p-1}}{p!} \|x_k - v_k\|^{p+1} + \sum_{i=1}^{p-1} \frac{\delta_i}{i!} \|x_k - v_k\|^{i+1}.$$

Thus,

$$\langle F(x_k), x_k - v_{k+1} \rangle$$

$$\leq \frac{L_{p-1}(1+5p)}{p!} \|x_k - v_k\|^{p} \|v_k - v_{k+1}\| - \frac{L_{p-1}(5p-2)}{p!} \|x_k - v_k\|^{p+1}$$

$$+ \sum_{i=1}^{p-1} \frac{\delta_i (1+\eta_i)}{i!} \|x_k - v_k\|^{i} \|v_k - v_{k+1}\| - \sum_{i=1}^{p-1} \frac{\delta_i (\eta_i - 2)}{i!} \|x_k - v_k\|^{i+1}$$

From this we get

$$\mathbf{II} = \sum_{k=1}^{T} \lambda_k \langle F(x_k), x_k - v_{k+1}\rangle - \tfrac{1}{2}\|v_k - v_{k+1}\|^2$$

$$\leq \sum_{k-1}^{T} \left[ \lambda_k \tfrac{L_{p-1}(1+5p)}{p!}\|x_k - v_k\|^p \|v_k - v_{k+1}\| - \tfrac{\lambda_k L_{p-1}(5p-2)}{p!}\|x_k - v_k\|^{p+1} \right.$$

$$+\lambda_k \sum_{i=1}^{p-1} \tfrac{\delta_i(1+\eta_i)}{i!}\|x_k - v_k\|^i \|v_k - v_{k+1}\| - \lambda_k \sum_{i=1}^{p-1} \tfrac{\delta_i(\eta_i-2)}{i!}\|x_k - v_k\|^{i+1} - \tfrac{1}{2}\|v_k - v_{k+1}\|^2 \Big]$$

$$\stackrel{\eta_i=5p}{=} \sum_{k=1}^{T} \left[ \lambda_k \left( \tfrac{L_{p-1}}{p!}\|x_k - v_k\|^{p-1} + \sum_{i=1}^{p-1} \tfrac{\delta_i}{i!}\|x_k - v_k\|^{i-1} \right)(5p+1)\|x_k - v_k\|\|v_k - v_{k+1}\| \right.$$

$$-\lambda_k \left( \tfrac{L_{p-1}}{p!}\|x_k - v_k\|^{p-1} + \sum_{i=1}^{p-1} \tfrac{\delta_i}{i!}\|x_k - v_k\|^{i-1} \right)(5p-2)\|x_k - v_k\|^2 - \tfrac{1}{2}\|v_k - v_{k+1}\|^2 \Big].$$

Now, if we choose $\lambda_k$ in a such way that

$$\tfrac{1}{4(5p-2)} \leq \lambda_k \left( \tfrac{L_{p-1}}{p!}\|x_k - v_k\|^{p-1} + \sum_{i=1}^{p-1} \tfrac{\delta_i}{i!}\|x_k - v_k\|^{i-1} \right) \leq \tfrac{1}{2(5p+1)}, \qquad (59)$$

we get the following

$$\mathbf{II} \stackrel{(59)}{\leq} \sum_{k=1}^{T} \left[ \tfrac{1}{2}\|x_k - v_k\|\|v_k - v_{k+1}\| - \tfrac{1}{4}\|x_k - v_k\|^2 - \tfrac{1}{2}\|v_k - v_{k+1}\|^2 \right]$$

$$\leq \sum_{k=1}^{T} \left[ \max_{\gamma} \left\{ \tfrac{1}{2}\gamma\|x_k - v_k\| - \tfrac{1}{2}\gamma^2 \right\} - \tfrac{1}{4}\|x_k - v_k\|^2 \right]$$

$$\leq \sum_{k=1}^{T} \left[ \tfrac{1}{4}\|x_k - v_k\|^2 - \tfrac{1}{8}\|x_k - v_k\|^2 - \tfrac{1}{4}\|x_k - v_k\|^2 \right]$$

$$= -\tfrac{1}{8}\sum_{k=1}^{T}\|x_k - v_k\|^2.$$

Finally, combining estimations of **I** and **II** with (57), we get

$$\sum_{t=1}^{T}\lambda_k \langle F(x_k), x_k - x\rangle \leq \mathcal{E}_0 - \mathcal{E}_T + \langle s_T, x - x_0\rangle - \tfrac{1}{8}\sum_{k=1}^{T}\|x_k - v_k\|^2.$$

$\square$

**Lemma G.4** *Let Assumptions 4.2, 1.3 hold. For every $T \geq 1$ we have*

$$\frac{1}{\sum_{k=1}^{T}\lambda_k} \leq \frac{2^p p^{\frac{p-1}{2}}(20p-8)\frac{L_{p-1}}{p!}\|x^* - x_0\|^{p-1}}{T^{\frac{p+1}{2}}} + 2^{\frac{3p}{2}}p^{\frac{p-1}{2}}(20p-8)\sum_{i=1}^{p-1}\frac{\delta_i\|x^* - x_0\|^{i-1}}{i!T^{\frac{i+1}{2}}}. \quad (60)$$

*Proof.* From Hölder inequality

$$\sum_{k=1}^{T}\lambda_k \geq \frac{T^{\frac{p+1}{2}}}{\left(\sum_{k=1}^{T}\lambda_k^{-\frac{2}{p-1}}\right)^{\frac{p-1}{2}}}$$

Consider denominator:

$$\sum_{k=1}^{T} \lambda_k^{-\frac{2}{p-1}} \overset{(59)}{\leq} \sum_{k=1}^{T} \lambda_k^{-\frac{2}{p-1}} (4(5p-2))^{\frac{2}{p-1}} \lambda_k^{\frac{2}{p-1}} \left( \frac{L_{p-1}}{p!} \|x_k - v_k\|^{p-1} + \sum_{i=1}^{p-1} \frac{\delta_i}{i!} \|x_k - v_K\|^{i-1} \right)^{\frac{2}{p-1}}$$

$$= \sum_{k=1}^{T} (20p-8)^{\frac{2}{p-1}} \left( \frac{L_{p-1}}{p!} \|x_k - v_k\|^{p-1} + \sum_{i=1}^{p-1} \frac{\delta_i}{i!} \|x_k - v_k\|^{i-1} \right)^{\frac{2}{p-1}}.$$

For $p \geq 2$ we have

$$\begin{cases} \frac{2}{p-1} = 2, & p = 2, \\ \frac{2}{p-1} \leq 1, & p \geq 3. \end{cases}$$

Consider some nonnegative sequence $\{a_i | a_i \geq 0, \ \forall i \in \overline{1,n}\}$. For $p = 2$ from Jensen inequality we know that $\left(\sum_{i=1}^{n} a_i\right)^2 \leq \sum_{i=1}^{n} n a_i^2$. From Lemma 7 of [65] we know that $\forall q \in [0,1], x, y > 0 \rightarrow (x+y)^q \leq x^q + y^q$. Thus, for $p \geq 3$ we can come to the same conclusion: $\left(\sum_{i=1}^{n} a_i\right)^{\frac{2}{p-1}} \leq \sum_{i=1}^{n} a_i^{\frac{2}{p-1}} \leq \sum_{i=1}^{n} n a_i^{\frac{2}{p-1}}$. In other words,

$$\left( \sum_{i=1}^{n} a_i \right)^{\frac{2}{p-1}} \leq \sum_{i=1}^{n} a_i^{\frac{2}{p-1}} \leq \sum_{i=1}^{n} n a_i^{\frac{2}{p-1}}, \quad \forall p \geq 2, a_i \geq 0. \tag{61}$$

Using this inequality, we get

$$\sum_{k=1}^{T} \lambda_k^{-\frac{2}{p-1}} \overset{(61)}{\leq} p(20p-8)^{\frac{2}{p-1}} \sum_{k=1}^{T} \left( \frac{L_{p-1}}{p!} \right)^{\frac{2}{p-1}} \|x_k - v_k\|^2 + p(20p-8)^{\frac{2}{p-1}} \sum_{k=1}^{T} \sum_{i=1}^{p-1} \left( \frac{\delta_i}{i!} \right)^{\frac{2}{p-1}} \|x_k - v_k\|^{\frac{2(i-1)}{p-1}}$$

$$\overset{(38)}{\leq} 4p(20p-8)^{\frac{2}{p-1}} \left( \frac{L_{p-1}}{p!} \right)^{\frac{2}{p-1}} \|x^* - x_0\|^2 + p(20p-8)^{\frac{2}{p-1}} \sum_{k=1}^{T} \sum_{i=1}^{p-1} \left( \frac{\delta_i}{i!} \right)^{\frac{2}{p-1}} \|x_k - v_k\|^{\frac{2(i-1)}{p-1}}$$

Consider the second factor in this inequality. If we denote $a_k = \|x_k - v_k\|^{\frac{2(i-1)}{p-1}}$, $b_k = 1$, $c = \frac{p-1}{i-1}$, $d = \frac{p-1}{p-i}$, then we can use Hölder inequality:

$$\sum_{k=1}^{n} |a_k b_k| \leq \left( \sum_{k=1}^{n} |a_k|^c \right)^{\frac{1}{c}} \left( \sum_{i=1}^{n} |b_k|^d \right)^{\frac{1}{d}}, \quad \frac{1}{c} + \frac{1}{d} = 1.$$

From this we get

$$\sum_{k=1}^{T} \|x_k - v_k\|^{\frac{2(i-1)}{p-1}} \leq \left( \sum_{k=1}^{T} \|x_k - v_k\|^2 \right)^{\frac{i-1}{p-1}} \left( \sum_{k=1}^{T} 1 \right)^{\frac{p-i}{p-1}}$$

$$\overset{(38)}{\leq} 4\|x^* - x_0\|^{\frac{2(i-1)}{p-1}} T^{\frac{p-i}{p-1}}.$$

Thus,

$$\sum_{k=1}^{T} \lambda_k^{-\frac{2}{p-1}} \leq 4p(20p-8)^{\frac{2}{p-1}} \left( \frac{L_{p-1}}{p!} \right)^{\frac{2}{p-1}} \|x^* - x_0\|^2 + 4p(20p-8)^{\frac{2}{p-1}} \sum_{i=1}^{p-1} \left( \frac{\delta_i}{i!} \right)^{\frac{2}{p-1}} \|x^* - x_0\|^{\frac{2(i-1)}{p-1}} T^{\frac{p-i}{p-1}}$$

Consider $(a+b)^{\frac{p-1}{2}}$, $p \geq 2$. For $p = 2$ from Lemma 7 in [65] we get $(a+b)^{\frac{1}{2}} \leq a^{\frac{1}{2}} + b^{\frac{1}{2}} < 2(a^{\frac{1}{2}} + b^{\frac{1}{2}})$. For $p \geq 3$ we have from Jensen inequality $(a+b)^{\frac{p-1}{2}} \leq 2^{\frac{p-1}{2}-1} \left( a^{\frac{p-1}{2}} + b^{\frac{p-1}{2}} \right) < 2^{\frac{p}{2}} \left( a^{\frac{p-1}{2}} + b^{\frac{p-1}{2}} \right)$. In other words,

$$(a+b)^{\frac{p-1}{2}} \leq 2^{\frac{p}{2}} \left( a^{\frac{p-1}{2}} + b^{\frac{p-1}{2}} \right), \quad \forall p \geq 2. \tag{62}$$

Thus,

$$\left(\sum_{k=1}^{T} \lambda_k^{-\frac{2}{p-1}}\right)^{\frac{p-1}{2}}$$

$$\overset{(62)}{\leq} 2^{\frac{p}{2}} \left[(4p)^{\frac{p-1}{2}}(20p-8)\frac{L_{p-1}}{p!}\|x^*-x_0\|^{p-1} + (4p)^{\frac{p-1}{2}}(20p-8)\left(\sum_{i=1}^{p-1}\left(\frac{\delta_i}{i!}\right)^{\frac{2}{p-1}}\|x^*-x_0\|^{\frac{2(i-1)}{p-1}}T^{\frac{p-i}{p-1}}\right)^{\frac{p-1}{2}}\right]$$

$$\leq 2^p p^{\frac{p-1}{2}}(20p-8)\left[\frac{L_{p-1}}{p!}\|x^*-x_0\|^{p-1} + 2^{\frac{p}{2}}\sum_{i=1}^{p-1}\frac{\delta_i}{i!}\|x^*-x_0\|^{i-1}T^{\frac{p-i}{2}}\right]$$

$$= 2^p p^{\frac{p-1}{2}}(20p-8)\frac{L_{p-1}}{p!}\|x^*-x_0\|^{p-1} + 2^{\frac{3p}{2}}p^{\frac{p-1}{2}}(20p-8)\sum_{i=1}^{p-1}\frac{\delta_i}{i!}\|x^*-x_0\|^{i-1}T^{\frac{p-i}{2}}.$$

Now, we can get the final result

$$\frac{1}{\sum_{k=1}^{T}\lambda_k} \leq \frac{\left(\sum_{k=1}^{T}\lambda_k^{-\frac{2}{p-1}}\right)^{\frac{p-1}{2}}}{T^{\frac{p+1}{2}}}$$

$$\leq \frac{2^p p^{\frac{p-1}{2}}(20p-8)\frac{L_{p-1}}{p!}\|x^*-x_0\|^{p-1}}{T^{\frac{p+1}{2}}} + 2^{\frac{3p}{2}}p^{\frac{p-1}{2}}(20p-8)\sum_{i=1}^{p-1}\frac{\delta_i\|x^*-x_0\|^{i-1}}{i!T^{\frac{i+1}{2}}}$$

$\square$

Finally, we provide the convergence rate of Algorithm 3 in monotone case.

**Theorem 7.2** *Let Assumptions 1.1, 4.2 with $i = p-1$, and 7.1 hold. Then, after $T \geq 1$ iterations of VIHI with parameters $\eta_i = 5p$, $\mathbf{opt} = 0$, we get the following bound*

$$\text{GAP}(\tilde{x}_T) = \sup_{x \in \mathcal{X}}\langle F(x), \tilde{x}_T - x\rangle \leq O\left(\frac{L_{p-1}D^{p+1}}{T^{\frac{p+1}{2}}} + \sum_{i=1}^{p-1}\frac{\delta_i D^{i+1}}{T^{\frac{i+1}{2}}}\right).$$

*Proof.* Consider $\forall x \in \mathcal{X}$, opt $= 0$.

$$\langle F(x), \tilde{x}_T - x\rangle$$

$$= \frac{1}{\sum_{k=1}^{T}\lambda_k}\sum_{k-1}^{T}\lambda_k\langle F(x_k), x_k - x\rangle \overset{(56)}{\leq} \frac{1}{\sum_{k=1}^{T}\lambda_k}\frac{1}{2}\|x-x_0\|^2 \leq \frac{D^2}{2\sum_{i=1}^{T}\lambda_k}$$

$$\overset{(60)}{\leq} 2^{p-1}p^{\frac{p-1}{2}}(20p-8)\frac{L_{p-1}D^{p+1}}{p!T^{\frac{p+1}{2}}} + 2^{\frac{3p}{2}-1}p^{\frac{p-1}{2}}(20p-8)\sum_{i=1}^{p-1}\frac{\delta_i D^{i+1}}{i!T^{\frac{i+1}{2}}}.$$

Thus,

$$\text{GAP}(\tilde{x}_T) = \sup_{x \in \mathcal{X}}\langle F(x), \tilde{x}_T - x\rangle = O\left(\frac{L_{p-1}D^{p+1}}{T^{\frac{p+1}{2}}} + \sum_{i=1}^{p-1}\frac{\delta_i D^{i+1}}{T^{\frac{i+1}{2}}}\right).$$

$\square$

### G.4   Convergence in nonmonotone case

To make our paper more self-contained, we provide convergence rate of Algorithm 3 in nonmonotone case.

**Theorem 7.3** *Let Assumptions 4.2, 1.3 and 7.1 with $i = p-1$ hold. Then after $T \geq 1$ iterations of Algorithm 3 with parameters $\eta = 5p$, opt $= 2$ we get the following bound*

$$\text{RES}(\hat{x}) := \sup_{x \in \mathcal{X}}\langle F(\hat{x}_t), \hat{x}_t - x\rangle = O\left(\frac{L_{p-1}D^{p+1}}{T^{\frac{p}{2}}} + \sum_{i=1}^{p-1}\frac{\delta_i D^{i+1}}{T^{\frac{i}{2}}}\right).$$

*Proof.* Consider $\langle F(x_k), x_k - x\rangle$.

$$\langle F(x_k), x_k - x\rangle = \langle F(x_k) - \Omega_{p,v_k}(x_k), x_k - x\rangle + \langle \Omega_{p,v_k}(x_k), x_k - x\rangle$$

$$\overset{(53)}{\leq} \|F(x_k) - \Omega_{p,v_k}(x_k)\|\|x_k - x\| + \frac{L_{p-1}}{p!}\|x_k - v_k\|^{p+1} + \sum_{i=1}^{p-1}\frac{\delta_i}{i!}\|x_k - v_k\|^{i+1}.$$

From definition of $\Psi_{p,v_k}(x_k)$ and triangle inequality we get

$$\|F(x_k) - \Psi_{p,v_k}(x_k)\| = \|F(x_k) - \Omega_{p,v_k}(x_k)\| - \frac{5L_{p-1}}{(p-1)!}\|x_k - v_k\|^p - \sum_{i=1}^{p-1}\frac{\eta_i\delta_i}{i!}\|x_k - v_k\|^i.$$

From this and (55) we get

$$\|F(x_k) - \Omega_{p,v_k}(x_k)\| \overset{(55)}{\leq} \frac{L_{p-1}}{p!}\|x_k - v_k\|^p + \sum_{i=1}^{p-1}\frac{1}{i!}\delta_i\|x_k - v_k\|^i + \frac{5L_{p-1}}{(p-1)!}\|x_k - v_k\|^p + \sum_{i=1}^{p-1}\frac{\eta_i\delta_i}{i!}\|x_k - v_k\|^i$$

$$= \frac{L_{p-1}}{p!}(5p+1)\|x_k - v_k\|^p + \sum_{i=1}^{p-1}\frac{\delta_i}{i!}(\eta_i + 1)\|x_k - v_k\|^i.$$

Thus,

$$\langle F(x_k), x_k - x\rangle$$

$$\leq \frac{L_{p-1}}{p!}(5p+1)\|x_k - v_k\|^p\|x_k - x\| + \sum_{i=1}^{p-1}\frac{\delta_i}{i!}(\eta_i + 1)\|x_k - v_k\|^i\|x_k - x\|$$

$$+ \frac{L_{p-1}}{p!}\|x_k - v_k\|^{p+1} + \sum_{i=1}^{p-1}\frac{\delta_i}{i!}\|x_k - v_k\|^{i+1}$$

$$\leq \frac{L_{p-1}}{p!}\|x_k - v_k\|^p(5p+2)D + \sum_{i=1}^{p-1}\frac{\delta_i}{i!}\|x_k - v_k\|^i(\eta_i + 2)D.$$

From second inequality of (38) and definition of $x_{k_T}$ we get

$$\|x_{k_T} - v_{k_t}\|^2 \equiv \min_{1\leq k\leq T}\|x_k - v_k\|^2 \leq \frac{1}{T}\sum_{k=1}^{T}\|x_k - v_k\|^2 \leq \frac{4}{T}\|x^* - x_0\|^2 \leq \frac{4D^2}{T}$$

$$\Leftrightarrow \|x_{k_T} - v_{k_T}\| \leq \frac{4^{\frac{i}{2}}D^i}{T^{\frac{i}{2}}}.$$

Combining all these results, we get

$$\text{RES}(\hat{x}) = \sup_{x\in\mathcal{X}}\langle F(x_k), x_k - x\rangle \leq \frac{L_{p-1}}{p!}\|x_k - v_k\|^p(5p+2)D + \sum_{i=1}^{p-1}\frac{\delta_i}{i!}\|x_k - v_k\|^i(5p+2)D$$

$$\leq \frac{2^p(5p+2)}{p!}\frac{L_{p-1}D^{p+1}}{T^{\frac{p}{2}}} + \sum_{i=1}^{p-1}\frac{2^i\delta_i}{i!}(\eta_i + 2)\frac{D^{i+1}}{T^{\frac{i}{2}}}$$

$$= O\left(\frac{L_{p-1}D^{p+1}}{T^{\frac{p}{2}}} + \sum_{i=1}^{p-1}\frac{\delta_i D^{i+1}}{T^{\frac{i}{2}}}\right)$$

$\square$

# H  Subproblem solution

For $r$-rank QN approximation $J^r$ from (19), we can effectively compute $A_\tau^{-1}F(x)$, where $A_\tau = J^r + (\eta\delta + \frac{5L}{2}\tau)I = U^\top CV + J^0 + (\eta\delta + \frac{5L}{2}\tau)I = U^\top CV + B$ and $B = J^0 + (\eta\delta + \frac{5L}{2}\tau)I$ by using the Woodbury matrix identity[96, 97].

$$A_\tau^{-1}F(x) = \left(B + U^\top CV\right)^{-1}F(x) = B^{-1}F(x) - B^{-1}U^\top(C^{-1} + VB^{-1}U^\top)^{-1}VB^{-1}F(x).$$

For $J_0 = \iota I$, the identity could be simplified even more. Hence, $B = \iota I + (\eta\delta + \frac{5L}{2}\tau)I = \chi I$, where $\chi = \iota + \eta\delta + \frac{5L}{2}\tau$. Then,

$$A_\tau^{-1}F(x) = \left(\chi I + U^\top CV\right)^{-1}F(x) = \frac{1}{\chi}F(x) - \frac{1}{\chi}U^\top(\chi C^{-1} + VU^\top)^{-1}VF(x).$$

# I  Experiment details

We consider the cubic regularized bilinear min-max problem of the form:

$$\min_{x\in\mathbb{R}^d}\max_{y\in\mathbb{R}^d} f(x,y) = y^\top(Ax - b) + \frac{\rho}{6}\|x\|^3, \tag{63}$$

where $\rho > 0$, $b = [1, 0, \ldots, 0] \in \mathbb{R}^d$, and $A \in \mathbb{R}^{d\times d}$ such that

$$A = \begin{bmatrix} 1 & -1 & 0 & \cdots & 0 \\ 0 & 1 & -1 & \ddots & \vdots \\ \vdots & \ddots & \ddots & \ddots & 0 \\ 0 & \cdots & 0 & 1 & -1 \\ 0 & \cdots & 0 & 0 & 1 \\ . & & & & \end{bmatrix}$$

To reformulate it as variational inequality, we define $F(x) = [\nabla_x f(x,y), -\nabla_y f(x,y)]$. Following [71], we plot the restricted primal-dual gap (65), written in a closed form:

$$\mathrm{gap}(z,\beta) = \frac{\rho}{6}\|x\|^3 + \beta\|Ax - b\| + \frac{2}{3}\sqrt{\frac{2}{\rho}}\|A^\top y\|^{\frac{3}{2}} + b^\top y,$$

where $z = (x,y)$.

**Setup.** All methods and experiments were performed using Python 3.11.5, PyTorch 2.1.2, numpy 1.24.3 on a 16-inch MacBook Pro 2023 with an Apple M2 Pro and 32GB memory.

**Parameters.** For Figure 1, the dimension of the problem $d = 50$. The regularizer $\rho$ is set to $\rho = 1e - 3$. The starting point $(x_0, y_0) = (0, 0)$ is all zeroes. We present results of last iteration of each method or opt=1 from Algorighm 1. The diameter $\beta$ for the gap is set as $\beta = 1$. For QN methods, we set memory size $m = r = 20$. We perform a total of 100000 iterations for all methods except Perseus2 with 1000 iterations. We plot each 500 iterations for a better visualisation. In Figure 1, the left plot refers to the gap decrease per iteration and the right plot refers to the gap decrease per JVP/operator computations. EG, Perseus1, VIQA computes 2 operators per iteration and Perseus2 computes $d + 1$ JVP/operators.

We finetuned learning rate and presented the run with the best results. For EG1: $lr = 0.5$. For Perseus1: $L_0 = 0.2334$. For Perseus2: $L_1 = 0.0001$. For VIQA Broyden: $L_1 = 0.001$, $\delta = J^0 = 0.4$. For VIQA Damped Broyden: $L_1 = 0.001$, $\delta = J^0 = 0.22$. Note, that tuned $L_1 = 0.001$ is actually a theoretical smoothness constant for the problem (63) and can be chosen with such value without finetuning. Also, Perseus2 is working with a smaller constant than theoretical.

# J  Application to minmax problems

## J.1  Preliminaries

In this section, we consider the problem of finding a global saddle point of the following min-max optimization problem:

$$\min_{x\in\mathbb{R}^m}\max_{y\in\mathbb{R}^n} f(x,y), \tag{64}$$

i.e., a tuple $(x^*, y^*) \in \mathbb{R}^m \times \mathbb{R}^n$ such that

$$f(x^*, y) \leq f(x^*, y^*) \leq f(x, y^*), \quad \text{for all } x \in \mathbb{R}^m, \ y \in \mathbb{R}^n,$$

where the continuously differentiable objective function $f$ is convex-concave: $f(x,y)$ is convex in $x$ for all $y \in \mathbb{R}^n$ and concave in $y$ for all $x \in \mathbb{R}^m$.

---

**Algorithm 4** VIJI-MinMax

---

**Input:** initial point $z_0 \in \mathcal{X}$, parameters $L_1, \delta, \eta, \tau$.
**Initialization:** set $s_0 = 0 \in \mathbb{R}^d$.
**for** $k = 0, 1, 2, \dots, T$ **do**
    Set $v_{k+1} = z_0 + s_{k+1}$.
    Compute $z_{k+1} \in \mathbb{R}^{n+m}$ such that condition (68) holds true.
    Compute $\lambda_{k+1}$ such that $\frac{1}{16} \leq \lambda_{k+1} \left( \frac{L}{2} \| z_{k+1} - v_{k+1} \| + \delta \right) \leq \frac{1}{12}$.
    Compute $s_{k+1} = s_k - \lambda_{k+1} F(z_{k+1})$.
**Output:** $\tilde{z}_T = \frac{1}{\sum_{k=1}^T \lambda_k} \sum_{k=1}^T \lambda_k z_k$.

---

**Assumption J.1** *The function $f(x, y) \in \mathcal{C}^2$ has L-Lipschitz-continious second-order derivative if*

$$\| \nabla^2 f(z) - \nabla^2(v) \| \leq L \| z - v \|, \quad \text{for all } z, v \in \mathbb{R}^{n+m}.$$

Following [71], we define the restricted gap function to measure the optimality of point $\hat{z} = (\hat{x}, \hat{y})$

$$\text{gap}(\hat{z}, \beta) = \max_{y: \| y - y^* \| \leq \beta} f(\hat{x}, y) - \min_{x: \| x - x^* \| \leq \beta} f(x, \hat{y}) \tag{65}$$

where $\beta$ is sufficiently large such that $\| \hat{z} - z^* \| \leq \beta$.

Problem (64) is a special case of VI defined by the following operator

$$F(\mathbf{z}) = \begin{bmatrix} \nabla_{\mathbf{x}} f(x, y) \\ -\nabla_{\mathbf{y}} f(x, y) \end{bmatrix}. \tag{66}$$

The Jacobian of $F$ is defined as follows

$$\nabla^2 F(z) = \begin{bmatrix} \nabla_{xx}^2 f(x, y) & \nabla_{xy}^2 f(x, y) \\ -\nabla_{xy}^2 f(x, y) & -\nabla_{yy}^2 f(x, y) \end{bmatrix}. \tag{67}$$

The following lemma [80, 71] provides properties of operator $F$.

**Lemma J.2** *Let Assumption J.1 hold. Then*

*1. The operator $F$ is monotone, i.e. satisfies Assumption 1.1.*
*2. The operator $F$ is L-smooth, i.e. satisfies Assumption 1.2.*
*3. $F(z^*) = 0$ for any global saddle point $z^* \in \mathbb{R}^{n+m}$ of the function $f$.*

We assume that inexact approximation of Jacobian satisfies Assumption 2.1.

## J.2   The method

Since we consider unconstrained minmax optimization, Step 2 of VIJI simplifies to $v_{k+1} = z_0 + s_k$ and the subproblem (11) changes to

$$\text{find } z_{k+1} \in \mathbb{R}^{n+m} \text{ such that } \Omega_{v_{k+1}}^\eta (z_{k+1}) = 0.$$

Usually, to find the solution of this subproblem it is necessary to run some addition subroutine. Following the work [71], we introduce the following approximate condition:

$$\| \Omega_{v_{k+1}}^\eta (z_{k+1}) \| \leq \tau \min \left\{ \frac{L}{2} \| z_{k+1} - v_{k+1} \|^2 + \delta \| z_{k+1} - v_{k+1} \|, \| F(v_{k+1}) \| \right\}, \tag{68}$$

where $\tau \in (0, 1)$ is a tolerance parameter.

The version of VIJI for unconstrained min-max problems is referred to as VIJI-MinMax and is detailed in Algorithm 4. This subproblem can solved by strategy proposed in [71], resulting in $O \left( (n+m)^\omega \varepsilon^{-2/3} + (n+m)^2 \varepsilon^{-2/3} \log \log(1/\varepsilon) \right)$ complexity, where $\omega \approx 2.3728$ is the matrix multiplication constant.

### J.3 Convergence analysis

The following theorem provides convergence rate for Algorithm 4.

**Theorem J.3** *Let Assumptions J.1, 2.1 hold. Then, after $T \geq 1$ iterations of VIJI-MinMax with parameters $\eta = 5$, we get the following bound*

$$\mathrm{gap}(\tilde{z}_T, \beta) \leq \frac{1152\sqrt{2}L\|z_0 - z^*\|^3}{T^{3/2}} + \frac{576\sqrt{2}\delta\|z_0 - \tilde{z}^*\|^2}{T},$$

*where $\beta = 5\|z_0 - z^*\|$, $z^* = (x^*, y^*)$.*

Now, let us introduce additional assumption on $\delta$, to obtain the convergence with optimal rate $O(\varepsilon^{-2/3})$.

**Theorem J.4** *Let Assumptions J.1 hold. Let*

$$\|(\nabla F(v_k) - J(v_k))[z_k - v_k]\| \leq \delta_k \|z_k - v_k\|$$

*hold for iterates $\{z_k, v_k\}$ generated by Algorithm 4, where*

$$\delta_k \leq L\|z_k - v_k\|. \tag{69}$$

*Then, after $T \geq 1$ iterations of VIJI-MinMax with parameters $\eta = 5$, we get the following bound*

$$\mathrm{gap}(\tilde{z}_T, \beta) \leq \frac{1944\sqrt{2}L\|z_0 - z^*\|^3}{T^{3/2}},$$

*where $\beta = 5\|z_0 - z^*\|$, $z^* = (x^*, y^*)$.*

Note, that it is also possible to change adaptive strategy for $\lambda_k$ in this case to $\frac{1}{16} \leq \frac{3}{2}L\|z_k - v_k\| \leq \frac{1}{12}$ (by introducing parameter $\beta$ as in Algorithm 1) to eliminate the dependence on $\delta$ from the step size while achieving the same convergence rate.

Finally, let us show, that we mathch the convergence of [71] under the same assumptions on Jacobian's inexactness and subproblem's solution.

**Assumption J.5 ([71])** *Let*

$$\|(\nabla F(v_k) - J(v_k))[z_k - v_k]\| \leq \delta_k \|z_k - v_k\|, \quad \|J(v_k)\| \leq \kappa$$

*hold for iterates $\{z_k, v_k\}$ generated by Algorithm 4, where*

$$\delta_k \leq \min\left\{\tau_0, \frac{L(1-\tau)}{4\kappa + 6L}\|F(v_k)\|\right\}, \tag{70}$$

*where $\tau_0 < \frac{L}{4}$.*

Next, we show, that (69) follows from (70).

Let us consider two cases. If $\|z_k - v_k\| \leq 1$, then from (70), we get $\delta_k \leq \tau_0 \|z_k - v_k\| \leq \frac{L}{2}\|z_k - v_k\|$. Otherwise, if $\|z_k - v_k\| \geq 1$, we obtain from (68)

$$\|F(v_k)\| - \|J(v_k)\|\|z_k - v_k\| - \delta\|z_k - v_k\| - 5L\|z_k - v_k\| \leq \|\nabla\Omega_{v_k}(z_k)\| \leq \tau\|F(v_k)\|.$$

Using our assumptions, we get

$$(1-\tau)\|F(v_k)\| \leq \kappa\|z_k - v_k\| + \delta\|z_k - v_k\| + 5L\|z_k - v_k\|^2 \leq (\kappa + \delta + 5L)\|z_k - v_k\|$$
$$\leq (\kappa + \tfrac{21}{4}L)\|z_k - v_k\|.$$

Next,

$$\delta \leq \frac{L(1-\tau)}{4\kappa + 6L}\|F(v_k)\| \leq L\|z_k - v_k\|.$$

Thus, the assumptions of Theorem J.4 hold, and Algorithm 4 achieves $O(\varepsilon^{-2/3})$ convergence.

### J.4 Proof of Theorem J.3

Again, as in the proof of Theorem 3.2, we introduce the Lyapunov function

$$\mathcal{E}_k = \max_{v \in \mathbb{R}^{n+m}} \langle s_k, v - z_0 \rangle - \tfrac{1}{2}\|v - z_0\|^2. \tag{71}$$

Note that the scalar product $\langle s_k, v - z_0 \rangle$ can be omitted (see the proof of [71, Theorems 3.1, 4.1]), which would also eliminate the somewhat redundant Step 2 of Algorithm 4. However, we chose to retain it, as it does not affect the method's performance. We retain

**Lemma J.6** *Let Assumptions J.1, 2.1 hold. Then, for every integer $T \geq 1$, we have*

$$\sum_{k=1}^{T} \lambda_k \langle F(z_k), z_k - z \rangle \leq \mathcal{E}_0 - \mathcal{E}_T + \langle s_T, z - z_0 \rangle - \tfrac{1}{8}\left(\sum_{k=1}^{T}\|z_k - v_k\|^2\right), \quad \text{for all } z \in \mathbb{R}^{n+m}.$$

*Proof.* Following the proof of Lemma B.1, we arrive at (35), where the proofs begin to slightly diverge due to the changes in the subproblem. At this juncture, we have:

$$\sum_{k=1}^{T} \lambda_k \langle F(z_k), z_k - z \rangle \leq \mathcal{E}_0 - \mathcal{E}_T + \langle s_T, z - z_0 \rangle + \underbrace{\sum_{k=1}^{T} \lambda_k \langle F(z_k), z_k - v_{k+1} \rangle - \tfrac{1}{2}\|v_k - v_{k+1}\|^2}_{\mathbf{II}}.$$
$$\tag{72}$$

Then,

$$\langle F(z_k), z_k - v_{k+1} \rangle$$
$$= \langle F(z_k) - \Omega_{v_k}^{\eta}(z_k) + \eta\delta(z_k - v_k) + 5L\|z_k - v_k\|(z_k - v_k), z_k - v_{k+1} \rangle$$
$$+ \langle \Omega_{v_k}^{\eta}(z_k), z_k - v_{k+1} \rangle - 5L\|z_k - v_k\|\langle z_k - v_k, z_k - v_{k+1}\rangle - \eta\delta\langle z_k - v_k, z_k - v_{k+1}\rangle$$
$$\overset{\text{Lem. (2.2), (68)}}{\leq} \frac{L}{2}\|z_k - v_k\|^2\|z_k - v_{k+1}\| + \delta\|z_k - v_k\|\|z_k - v_{k+1}\| + \frac{\tau L}{2}\|z_k - v_k\|^2\|z_k - v_{k+1}\|$$
$$+ \tau\delta\|z_k - v_k\|\|z_k - v_{k+1}\| - 5L\|z_k - v_k\|\langle z_k - v_k, z_k - v_{k+1}\rangle - \eta\delta\langle z_k - v_k, z_k - v_{k+1}\rangle$$

Next, using $\langle z_k - v_k, z_k - v_{k+1}\rangle \geq \|z_k - v_k\|^2 - \|z_k - v_k\|\|v_k - v_{k+1}\|$ and $\|z_k - v_{k+1}\| \leq \|z_k - v_k\| + \|v_k - v_{k+1}\|$, we get

$$\langle F(z_k), z_k - v_{k+1} \rangle$$
$$\leq (\tau+1)\frac{L}{2}\|z_k - v_k\|^3 + (\tau+1)\frac{L}{2}\|z_k - v_k\|^2\|v_k - v_{k+1}\| + (\tau+1)\delta\|z_k - v_k\|^2$$
$$+ (\tau+1)\delta\|z_k - v_k\|\|v_k - v_{k+1}\| - 5L\|z_k - v_k\|\langle z_k - v_k, z_k - v_{k+1}\rangle - \eta\delta\langle z_k - v_k, z_k - v_{k+1}\rangle$$
$$\overset{\tau \leq 1}{\leq} 6L\|z_k - v_k\|^2\|v_k - v_{k+1}\| - 4L\|z_k - v_k\|^3$$
$$+ (\eta+1)\delta\|z_k - v_k\|\|v_k - v_{k+1}\| - (\eta-1)\delta\|z_k - v_k\|^2$$

Next,

$$\mathbf{II} \leq \sum_{k=1}^{T}\left(\tfrac{1}{2}\|z_k - v_k\|\|v_k - v_{k+1}\| - \tfrac{1}{4}\|z_k - v_k\|^2 - \tfrac{1}{2}\|v_k - v_{k+1}\|^2\right),$$

where the last inequality is due to the following choice of $\eta = 10$ and $\lambda$ :
$\tfrac{1}{16} \leq \lambda_k\left(L\|z_k - v_k\| + \delta\right) \leq \tfrac{1}{12}$. Then,

$$\mathbf{II} \leq \sum_{k=1}^{T} \tfrac{1}{2}\|z_k - v_k\|\|v_k - v_{k+1}\| - \tfrac{1}{4}\|z_k - v_k\|^2 - \tfrac{1}{2}\|v_k - v_{k+1}\|^2 \leq -\tfrac{1}{8}\left(\sum_{k=1}^{T}\|z_k - v_k\|^2\right).$$
$$\tag{73}$$

Plugging (73) into (72) yields that

$$\sum_{k=1}^{T} \lambda_k \langle F(z_k), z_k - z \rangle \leq \mathcal{E}_0 - \mathcal{E}_T + \langle s_T, z - z_0 \rangle - \tfrac{1}{8}\left(\sum_{k=1}^{T}\|z_k - v_k\|^2\right).$$

$\square$

Next, Lemma B.2 can be applied. Next Lemma is a counterpart of Lemma B.3 for the current choice of $\lambda$.

**Lemma J.7** *Let Assumptions J.1, 2.1 hold. For every integer $T \geq 1$, we have*

$$\frac{1}{\left(\sum_{k=1}^{T} \lambda_k\right)^2} \leq \frac{2048 L^2 \|z^* - z_0\|^2}{T^3} + \frac{512\delta^2}{T^2}, \tag{74}$$

*where $x^* \in \mathcal{X}$ denotes the weak solution to the VI.*

*Proof.* Without loss of generality, we assume that $z_0 \neq z^*$. We have

$$\sum_{k=1}^{T} (\lambda_k)^{-2} 16^{-2} \leq \sum_{k=1}^{T} (\lambda_k)^{-2} \left(\lambda_k \left(L \|z_k - v_k\| + \delta\right)\right)^2 = \sum_{k=1}^{T} \left(L \|z_k - v_k\| + \delta\right)^2$$

$$\leq \sum_{k=1}^{T} 2L^2 \|z_k - v_k\|^2 + 2T\delta^2 \overset{\text{Lemma B.2}}{\leq} 8L^2 \|z^* - z_0\|^2 + 2T\delta^2.$$

By the Hölder inequality, we have

$$\sum_{k=1}^{T} 1 = \sum_{k=1}^{T} \left((\lambda_k)^{-2}\right)^{1/3} (\lambda_k)^{2/3} \leq \left(\sum_{k=1}^{T} (\lambda_k)^{-2}\right)^{1/3} \left(\sum_{k=1}^{T} \lambda_k\right)^{2/3}.$$

Putting these pieces together yields that

$$T \leq 16^{2/3} (8L^2 \|z^* - z_0\|^2 + 2\delta^2 T)^{\frac{1}{3}} \left(\sum_{k=1}^{T} \lambda_k\right)^{2/3},$$

Plugging this into the above inequality yields that

$$\frac{1}{\left(\sum_{k=1}^{T} \lambda_k\right)^2} \leq \frac{2048 L^2 \|z^* - z_0\|^2}{T^3} + \frac{512\delta^2}{T^2}$$

$\square$

Next, by Lemma J.6, we have

$$0 \overset{\text{Lem. J.2}}{\leq} \sum_{k=1}^{T} \lambda_k \langle F(z_k), z_k - z \rangle + -\frac{1}{8} \left(\sum_{k=1}^{T} \|z_k - v_k\|^2\right) \leq \mathcal{E}_0 - \mathcal{E}_T + \langle s_T, z - z_0 \rangle$$

$$\overset{(71)}{=} \frac{1}{2} \|v_0 - z_0\|^2 - \frac{1}{2} \|v_T - z_0\|^2 + \langle v_T - z_0, v^* - v_0 \rangle.$$

By applying Young's ineqaulity and the fact that $z_0 = v_0$, we get

$$0 \leq -\frac{1}{2} \|v_k - z_0\|^2 + \frac{1}{4} \|v_k - z_0\|^2 + \|z^* - z_0\|^2 = -\frac{1}{4} \|v_k - z_0\|^2 + \|z^* - z_0\|^2.$$

Thus, we have $\|v_k - z_0\| \leq 2\|z^* - z_0\|$. From Lemma B.2 we also have that $\|z_k - v_k\| \leq 2\|z^* - z_0\|$. Thus,

$$\|v_k - z^*\| \leq \|v_k - z_0\| + \|z_0 - z^*\| \leq 3\|z_0 - z^*\| \leq \beta,$$
$$\|z_k - z^*\| \leq \|z_k - v_k\| + \|z_k - z^*\| \leq 5\|z_0 - z^*\| = \beta.$$

Lemma B.2 also implies

$$\sum_{k=1}^{T} \lambda_k (z_k - z)^\top F(z_k) \leq \tfrac{1}{2} \|z_0 - z\|^2.$$

By Proposition [71, Proposition 2.9], we have

$$f(\tilde{x}_T, y) - f(x, \tilde{y}_T) \leq \frac{1}{\sum_{k=1}^{T} \lambda_k} \left( \sum_{k=1}^{T} \lambda_k (z_k - z)^\top F(z_k) \right).$$

Putting these pieces together yields

$$f(\tilde{x}_T, y) - f(x, \tilde{y}_T) \leq \frac{1}{2(\sum_{k=1}^{T} \lambda_k)} \|z_0 - z\|^2.$$

This together with Lemma J.7 yields

$$f(\tilde{x}_T, y) - f(x, \tilde{y}_T) \leq \frac{8\sqrt{2}L \|z_0 - z^*\| \|z_0 - z\|^2}{T^{3/2}} + \frac{4\sqrt{2}\delta \|z_0 - z\|^2}{T}.$$

Since $\|z_k - z^*\| \leq \beta$ for all $k \geq 0$, we have $\|\tilde{z}_T - z^*\| \leq \beta$. By the definition of the restricted gap function, we have

$$\mathrm{gap}(\tilde{z}_T, \beta) \leq \frac{32\sqrt{2}L \|z_0 - z^*\| (\|z_0 - z^*\| + \beta)^2}{T^{3/2}} + \frac{16\sqrt{2}\delta (\|z_0 - z^*\| + \beta)^2}{T}$$

$$\leq \frac{1152\sqrt{2}L \|z_0 - z^*\|^3}{T^{3/2}} + \frac{576\sqrt{2}\delta \|z_0 - \tilde{z}^*\|^2}{T}.$$

Therefore, we conclude from the above inequality that there exists some $T > 0$ such that the output $\hat{z}$ satisfies that $\mathrm{gap}(\hat{z}, \beta) \leq \epsilon$.

## J.5 Proof of Theorem J.4

Lemma (J.6) hold with slight modification in $\lambda_k$ adaptive strategy. Lemma B.2 also holds. Next we need slight modification of Lemma B.3 for the current choice of $\lambda$.

**Lemma J.8** *Let Assumption J.1. For every integer $T \geq 1$, we have*

$$\frac{1}{\left( \sum_{k=1}^{T} \lambda_k \right)^2} \leq \frac{54L^2 \|z^* - z_0\|^2}{T^3},$$

*where $x^* \in \mathcal{X}$ denotes the weak solution to the VI.*

*Proof.* Without loss of generality, we assume that $z_0 \neq z^*$. We have

$$\sum_{k=1}^{T} (\lambda_k)^{-2} 16^{-2} \leq \sum_{k=1}^{T} (\lambda_k)^{-2} \left( \lambda_k \left( L \|z_k - v_k\| + \delta \right) \right)^2 = \sum_{k=1}^{T} \left( L \|z_k - v_k\| + \delta \right)^2$$

$$\overset{(69)}{\leq} \sum_{k=1}^{T} \frac{9L^2}{4} \|z_k - v_k\|^2 \overset{\text{Lemma B.2}}{\leq} \frac{9L^2}{4} \|z^* - z_0\|^2.$$

By the Hölder inequality, we have

$$\sum_{k=1}^{T} 1 = \sum_{k=1}^{T} \left( (\lambda_k)^{-2} \right)^{1/3} (\lambda_k)^{2/3} \leq \left( \sum_{k=1}^{T} (\lambda_k)^{-2} \right)^{1/3} \left( \sum_{k=1}^{T} \lambda_k \right)^{2/3}.$$

Putting these pieces together yields that

$$T \leq 16^{2/3} \left( \tfrac{9}{4} L^2 \|z^* - z_0\|^2 \right)^{\frac{1}{3}} \left( \sum_{k=1}^{T} \lambda_k \right)^{2/3},$$

Plugging this into the above inequality yields that

$$\frac{1}{\left( \sum_{k=1}^{T} \lambda_k \right)^2} \leq \frac{54L^2 \|z^* - z_0\|^2}{T^3}.$$

$\square$

Next, directly following the steps of proof of Theorem J.3, we get the following rate

$$\mathrm{gap}(\tilde{z}_T, \beta) \leq \frac{1944\sqrt{2}L \|z_0 - z^*\|^3}{T^{3/2}}.$$

