# OpenReview forum: "Exploring Jacobian Inexactness in Second-Order Methods for Variational Inequalities: Lower Bounds, Optimal Algorithms and Quasi-Newton Approximations"
_NeurIPS.cc/2024/Conference — NeurIPS 2024 spotlight_

### Official Review · Reviewer_Vnro · 2024-07-09

**Soundness:** 4
**Presentation:** 4
**Contribution:** 3
**Rating:** 8
**Confidence:** 4

**Summary:**

The paper develops 2nd-order methods for variational inequalities under inexact Jacobian information. Existing methods requires exact derivative information which can be prohibitively expensive for large-problems limiting their practicality. To address this issue, the paper analyzes the impact of inexact derivatives on the convergence rate. When the objective is smooth and monotone, the paper establishes a lower-bound that depends upon the level inaccuracy and develop an optimal algorithm that attains this bound. The paper develops several inexact schemes based on a Quasi-Newton approximation that achieve a gloabl sublinear convergence rates.

**Strengths:**

1.) The paper is the first to systematically explore the effect the impact of inexact Jacobian information on the global convergence rate of 2nd-order methods for variational inequalities. In particular, the established bounds demonstrate the precise impact of the level inexactness on the speed of convergence.

2.) In the important special case when F is smooth and monotone, the paper provides a lower-bound on the convergence rate that explicitly depends upon the level of inexactness. They then provide an algorithm that achieves this lower-bound. Thus, the analysis in the paper is quite tight.

3.) On a practical level the authors develop inexact 2nd-order schemes that achieve global sublinear convergence based on quasi-Newton schemes.

4.) The theory is supported by numerical simulation and comparision with existing methods.

5.) The paper is clear and easy to read.

Overall, I think the paper provides a strong contribution to optimization theory and should be accepted.

**Weaknesses:**

The main weakness that I can see is that the paper focuses on deterministic inaccuracy in the Jacobian. Most practical ML algorithms use stochastic derivative information. Moreover, the batch size doesn't typically increase as the algorithm progresses, so generally only a noisy neighborhood of the optimum is reached. Thus, it is not immediately clear how relevant the algorithms and analysis in the paper are to this setting.

**Questions:**

1) If the algorithms in this paper only have access to stochastic Jacobian information  and assuming they don't minibatch, do the authors believe the conclusions of the paper still hold modulo the fact that convergence will only occur up to the noise level in the Jacobian? If the answer is yes, then I don't think the weakness above is that significant.

2)  For the experiments results in Figure 1, how well do the proposed methods perform in terms of wall-clock time? Although they require more iterations then Perseus, I would expect them to be faster due to their use of inexactness. I think inclusion of this information in a table would make a nice addition to the paper.

**Limitations:**

I think the authors have addressed the main limitations of the work at the end of the paper.

---

> ### Author Rebuttal · Authors · 2024-08-07
>
> Dear Reviewer Vrno,
>
> Thank you for your detailed and thoughtful review of our paper. We are pleased to hear that you found our theoretical analysis and practical developments compelling. Your positive feedback on the clarity and readability of the writing is greatly appreciated. We also value your insights regarding stochastic Jacobians. Below, we address your questions in detail.
>
> **Weakness 1**
>
> >  The main weakness that I can see is that the paper focuses on deterministic inaccuracy in the Jacobian. Most practical ML algorithms use stochastic derivative information. Moreover, the batch size doesn't typically increase as the algorithm progresses, so generally only a noisy neighborhood of the optimum is reached. Thus, it is not immediately clear how relevant the algorithms and analysis in the paper are to this setting.
>
> **Question 1**
>
> >  If the algorithms in this paper only have access to stochastic Jacobian information and assuming they don't minibatch, do the authors believe the conclusions of the paper still hold modulo the fact that convergence will only occur up to the noise level in the Jacobian? If the answer is yes, then I don't think the weakness above is that significant.
>
> **Response to Weakness 1 and Question 1**
>
> Thank you for your insightful commentary. The application of our method to stochastic optimization remains an open question. To the best of our knowledge, there is not even a stochastic version of dual extrapolation, on which our method is based.
> The problem of applying the current scheme to stochastic problems arises in Lemma B1, where we do not know how to apply expectation to the difference between the Jacobian and its approximation. We calculate the Jacobian at the point $v_k$, while in lines 588-589, where we want to apply expectation, we have terms $v_{k+1}$ and $x_k$, which depend on stochastisity from stochastic Jacobian in $v_k$. Thus, we leave true stochasticity for future work.
> What one can do is to show the convergence of the method with high probability if Assumption 2.1 is satisfied with high probability. For example, the mini-batch scenario can be included here by using concentration inequalities. This analysis might not require increasing the batch size during iterations, but it usually leads to very large batch sizes.
>
> **Question 2**
>
> > For the experiments results in Figure 1, how well do the proposed methods perform in terms of wall-clock time? Although they require more iterations then Perseus, I would expect them to be faster due to their use of inexactness. I think inclusion of this information in a table would make a nice addition to the paper.
>
> **Response**
>
> Thank you for your suggestion. You are correct that the exact second-order Perseus2 step takes more time than an iteration of VIQA. However, we did not include a time comparison because we specifically chose this example to allow for fast computation of the operator and the Jacobian due to the problem's structure, aiming to challenge the convergence of first-order methods in terms of iteration complexity. To demonstrate the benefits in wall-clock time, we will need to use more complex problems. Our ongoing experiments are addressing this, and we plan to include relevant plots in the final version of the paper.

---

> > ### Comment · Reviewer_Vnro · 2024-08-07
> > **Thanks for the rebuttal**
> >
> > I appreciate the authors' response, in particular to their thoughtful answer to my question about stochastic Jacobians. I agree this sounds like an interesting direction for future work. Overall, the rebuttal has addressed my questions/concerns, I'm raising my score from 7 to 8.

---

> > > ### Author Response · Authors · 2024-08-11
> > >
> > > Thank you for your prompt response and for raising the score. We are pleased to hear that our rebuttal addressed your concerns.

---

### Official Review · Reviewer_jz9w · 2024-07-12

**Soundness:** 3
**Presentation:** 3
**Contribution:** 3
**Rating:** 6
**Confidence:** 4

**Summary:**

This paper studies the impact of inexactness Jacobian in second-order methods for solving variational inequalities.

This article is very informative， including a novel inexact second-order framework which achieves a convergence rate that matches the proposed lower bound, feasible solution to the sub problem, combination of quasi-Newton methods, and the generalization to the higher-order inexactness.

Overall, I found this paper well-written and the results are neat and  interesting.

**Strengths:**

1. This paper gives a novel second-order methods that allows the inexactness of the Jacobian. The convergence rate of the proposed methods  matches the lower bound given by the authors, which is a nice and neat result.

2. This paper involves solving sub VI problem, and implementation to quasi-Newton methods, which is practical.

3. This paper is generally well-written.

**Weaknesses:**

There may lack some discussion on the related work:

In terms of the inexact Jacobian: It is very necessary to discuss the relation with one that considers the inexact Jacobian for convex optimization, especially the technical difficulties over them [3, 4].

In terms of the quasi-Newton methods for solving VI, the author may need to discuss [A] which does not require the symmetric QN update for saddle point problems.

I will also be happy to see discussion on the **technical difficulties** in comparison with optimal second-order methods for solving VI, especially [59], where Algorithm 1 use very similar framework to.


[A]. Liu C, Luo L. Quasi-Newton methods for saddle point problems. NeurIPS 2022.

**Questions:**

See weakness.

In related work for convex optimization [3, 4], they also show the influence of inexact gradient on the convergence, it seems also important to discuss the behavior when $F(\cdot)$ is inexact.

---

> ### Author Rebuttal · Authors · 2024-08-05
>
> Dear Reviewer jz9w,
>
> Thank you for your thorough and insightful review of our paper! We are grateful for your recognition of the strengths of our work. We greatly appreciate your constructive feedback on areas needing further discussion. Below, we try to address the related works that prompted your questions. We will adjust the content in the paper accordingly.
>
> **Weakness 1**
> > In terms of the inexact Jacobian: It is very necessary to discuss the relation with one that considers the inexact Jacobian for convex optimization, especially the technical difficulties over them [3, 4].
>
> **Response**
>
> 1. *The nature of algorithms*. The methods in [3, 4] are inexact versions of  Cubic Newton, while our algorithm is based on Perseus, a high-order generalization of Nesterov’s dual extrapolation. This results in completely different theoretical analyses. Additionally, the minimization algorithms in [3, 4] are not optimal for the case of exact derivatives, whereas our VI algorithm is optimal.
> 2. *Subproblem solving*. While the methods in [3, 4] have minimization problems as subproblems, our method has a VI subproblem. This necessitates the use of different subroutines to solve these subproblems.
> 3. *Inaccuracy in Jacobians (Hessians)*. We use the same assumptions regarding the inexactness of the Jacobian/Hessian. Similar to [3, 4], we have included an additional regularization term inside the model (eq. 10). In the case of minimization, this term makes the objective's model convex and majorizes the objective. For VIs, this term is crucial for ensuring that the method’s subproblem has a solution.
> 4. *Symmetric vs. non-symmetric approximations*. The Jacobian is not necessarily a symmetric matrix, unlike the Hessian, leading to different approximation techniques. It might not be optimal to apply symmetric QN updates, such as SR1 and LBFGS, which are designed for minimization, to VI problems.
>
> **Weakness 2**
>
> >  In terms of the quasi-Newton methods for solving VI, the author may need to discuss [A] which does not require the symmetric QN update for saddle point problems.
>
> **Response**
>
> Thank you for pointing out this work. We will add the citation to our paper. Here are a few differences between our approaches:
> 1. [A] considers only min-max problems, not general VI.
> 2. [A] focuses on local convergence, whereas our approach addresses global convergence.
> 3. [A] uses symmetric QN approximations for the square of the Jacobian, while we construct truly non-symmetric QN approximations.
>
> **Weakness 3**
> >  I will also be happy to see discussion on the technical difficulties in comparison with optimal second-order methods for solving VI, especially [59], where Algorithm 1 use very similar framework to.
>
> **Response**
>
> 1. Model. To ensure that the method’s subproblem has a solution in the case of an inexact Jacobian, we have to construct a different model of the objective. To tackle this problem (Lemma 3.1), we introduce an additional regularization term  $\eta \delta (x-v)$  in the model of the objective (eq. 10).
>
> 2. The strategy chosen in Perseus [59] might not be suitable for the case of an inexact Jacobian, as it can lead to aggressive steps, slowing down the method. Thus, in our approach, the level of inexactness also affects the strategy for selecting adaptive step sizes in the dual space $\lambda_k$.
>
> 3. Theoretical analysis.
> For simplicity, let us start with second-order methods. We use the same Lyapunov function as in [59]. Due to the inexactness in the Jacobian, the bound on the difference between the objective and its model depends on the inexactness level $\delta$ (Lemma 2.2) as well as a criterion on the subproblem's solution (12). This leads to different analyses in Lemma B.1 and the necessity of a different strategy for $\lambda_k$ (line 589). Since we’ve changed the rule for $\lambda_k$, the bound on the sum of step sizes (Lemma B.3) changes as well, which finally leads to a convergence rate with explicit dependence on $\delta$.
> For the case of high-order methods, we again change the criteria for the subproblem's solution (eq. 53) and the step size policy (Algorithm 3): they depend on all inaccuracy levels $\delta_i$. This leads to Lemma G4, where we use Hölder and Jensen inequalities to derive the bound on the sum of step sizes.
>
> **Question 1**
>
> > In related work for convex optimization [3, 4], they also show the influence of inexact gradient on the convergence, it seems also important to discuss the behavior when F(⋅) is inexact.
>
> Thank you for your insightful commentary. Indeed, the dependence of the convergence on inexactness in the operator itself is an interesting open problem, which lies outside the scope of this work. We leave it for future research, as even stochastic dual extrapolation and dual extrapolation with inexact operators have not been studied in the literature yet.

---

> > ### Comment · Reviewer_jz9w · 2024-08-10
> >
> > I thank the authors for their response and have no further questions.

---

> > > ### Author Response · Authors · 2024-08-11
> > >
> > > Thank you for your response. We appreciate your feedback and are glad that we could address your concerns.

---

### Official Review · Reviewer_7Fy2 · 2024-07-13

**Soundness:** 3
**Presentation:** 3
**Contribution:** 3
**Rating:** 6
**Confidence:** 3

**Summary:**

This paper presents a new second-order algorithm for variational inequalities that is applicable to inexact Jacobian. The complete convergence analysis and a theoretical lower bound are provided in the paper.

**Strengths:**

1. A novel second-order algorithm is proposed for variational inequalities that is able to deal with inexact Jacobian.
2. Solid convergence analysis and a theoretical lower bounded are established.

**Weaknesses:**

The advantages of the proposed method over some related works are not stated clearly (see question 1).

**Questions:**

1. What is the advantage of the proposed method over [60] since the convergence rate is not improved? To my understanding, the assumption of Jacobian inexactness in [60] is not suitable for quasi-newton. But why "suitable to quasi-newton" can be claimed as an advantage?
2. Will inexact Jacobian causes more iterations to converge than exact Jacobian, which eventually leads to a higher complexity than exact Jacobian?

**Limitations:**

I did not see any negative societal impact.

---

> ### Author Rebuttal · Authors · 2024-08-05
>
> Dear Reviewer 7Fy2,
>
> Thank you for your valuable feedback! We appreciate your acknowledgment of the strengths of our work, including the tightness of the proposed upper and lower bounds. We also value your constructive feedback regarding areas that need further discussion, particularly in relation to related work. Below we answer your questions, and we will improve the discussion in the paper accordingly.
>
> > **Question 1** What is the advantage of the proposed method over [60] since the convergence rate is not improved? To my understanding, the assumption of Jacobian inexactness in [60] is not suitable for quasi-newton. But why "suitable to quasi-newton" can be claimed as an advantage?
>
> **Response**
>
> Indeed, when the inexactness level $\delta$ is controllable, the rate $O(\varepsilon^{-2/3})$ is not improved. However, $\Omega(\varepsilon^{-2/3})$ is the lower bound for exact algorithms, meaning any exact or inexact second-order algorithm cannot exceed this rate. Our algorithm can converge with an *arbitrary* inexactness level $\delta \geq 0$. Unlike [60], we do not need additional assumptions on the Jacobian's inexactness, allowing us to apply any approximation strategy satisfying $||J - \nabla F|| \leq \delta$. While [60] works only with controllable inexactness, so the accuracy of Jacobian approximation on last iterations is necessarily very high.
> One example is the Quasi-Newton (QN) update, which is widely used in modern minimization due to its practicality. Since it is typically hard to estimate the convergence of QN updates to the true Jacobian, the inexactness assumption proposed in [60] may not hold. Our algorithm can run with the worst-case estimation of the inexactness level of the QN update (Theorem 5.1), resulting in $O(\varepsilon^{-1})$  global convergence and significantly reduced iteration cost (Section 5). Thus, we got a globally convergent QN algorithm for VIs that, in practice, outperforms first-order methods using only first-order information. In summary, the advantages of QN updates include cheap iterations, theoretical global convergence, and practical performance.
> Another difference with [60] is that our method is applicable to general VIs, not only min-max problems.
> We will improve the discussion at the end of Section 1.
>
> > **Question 2** Will inexact Jacobian causes more iterations to converge than exact Jacobian, which eventually leads to a higher complexity than exact Jacobian?
>
> **Response**
>
> Yes, theoretically, we obtain the rate $O(\tfrac{\delta D^2}{T} + \tfrac{L_1D^3}{T^{3/2}})$, which matches the optimal convergence rate of second-order methods when $\delta \leq \tfrac{L_1 D}{\sqrt{T}}$​. In practice, we observe that the total number of iterations for the second-order Perseus is smaller than that for the inexact QN algorithm VIQA. However, computing the Jacobian and solving the second-order subproblem in each iteration can be expensive for exact second-order methods, especially in high-dimensional settings. This is where our proposed approach benefits.

---

### Decision · Program_Chairs · 2024-09-25

**Decision:**

Accept (spotlight)

**Comment:**

The authors present second-order methods for variational inequalities that can handle inexact Jacobian information, along with lower bounds that depend on the amount of inexactness. The reviewers all agree that the work is interesting and impactful, and thereby worthy of acceptance.